# Live imaging screen reveals that TYRO3 and GAK ensure accurate spindle positioning in human cells

Benita Wolf[1], Coralie Busso[1] & Pierre Gönczy[1]

Proper spindle positioning is crucial for spatial cell division control. Spindle positioning in human cells relies on a ternary complex comprising Gαi1–3, LGN and NuMA, which anchors dynein at the cell cortex, thus enabling pulling forces to be exerted on astral microtubules. We develop a live imaging siRNA-based screen using stereotyped fibronectin micropatterns to uncover components modulating spindle positioning in human cells, testing 1280 genes, including all kinases and phosphatases. We thus discover 16 components whose inactivation dramatically perturbs spindle positioning, including tyrosine receptor kinase 3 (TYRO3) and cyclin G associated kinase (GAK). TYRO3 depletion results in excess NuMA and dynein at the cortex during metaphase, similar to the effect of blocking the TYRO3 downstream target phosphatidylinositol 3-kinase (PI3K). Furthermore, depletion of GAK leads to impaired astral microtubules, similar to the effect of downregulating the GAK-interactor Clathrin. Overall, our work uncovers components and mechanisms governing spindle positioning in human cells.

[1] Swiss Institute for Experimental Cancer Research (ISREC), School of Life Sciences, Swiss Federal Institute of Technology Lausanne (EPFL), 1015 Lausanne, Switzerland. Correspondence and requests for materials should be addressed to P.G. (email: pierre.gonczy@epfl.ch)

Spatial control of cell division is crucial during development and for tissue homeostasis[1,2]. In animal cells, the position of the mitotic spindle can dictate the correct segregation of fate determinants during cell division, as well as the accurate placement of daughter cells within tissues and organisms. Defective spindle positioning in the vertebrate central nervous system can lead to tissue hyperplasia[3], indicative of accurate spindle positioning playing a tumor suppressive function[4]. Overall, proper spindle positioning is critical for the spatial regulation of cell division, and an understanding of the underlying mechanisms has important implications for proliferation control.

Astral microtubules that emanate from the two poles of the mitotic spindle and polymerize toward the cell cortex are key for ensuring proper spindle positioning (reviewed in ref. [5]). A further critical element is the cortically anchored minus-end directed motor dynein, which, together with depolymerizing astral microtubules, generates pulling forces that result in accurate spindle positioning (reviewed in ref. [6]). Dynein is anchored at the cell cortex through a ternary complex that comprises notably a large coiled coil protein called NuMA in human cells, LIN-5 in *Caenorhabditis elegans* and Mud in *Drosophila*[7–9]. In metaphase human cells, NuMA is anchored at the cell cortex through an interaction with the GoLoco-containing protein LGN, which itself is bound to the plasma membrane through an interaction with a Gαi protein[8]. At this stage of the cell cycle, most of the NuMA protein is phosphorylated by CDK1[6,10]. Given that this kinase is enriched at centrosomes, phosphorylated NuMa localizes primarily at spindle poles during metaphase[6]. Such phosphorylation is counteracted by PPP2CA, which is thought to act throughout the cell, together resulting in a low level of dephosphorylated NuMA at the cell cortex[6,11]. Upon loss of CDK1 activity in anaphase, NuMA binds to the cortex independently of LGN through an interaction with phosphatidylinositol 4,5-bisphosphate (PIP$_2$)[12,13]. Moreover, 4.1 band proteins may contribute to NuMA cortical anchoring in anaphase[14], although there are contrasting results on this aspect[12].

Regulation of spindle positioning, in particular through alterations of NuMA distribution, relies also on kinases other than CDK1. Thus, Aurora Kinase A phosphorylates NuMA during metaphase, promoting NuMA cortical enrichment and proper spindle positioning[15,16]. Accordingly, interfering with Aurora A function in the mammary epithelium impairs spindle position, resulting in altered cell fate and proliferation[17]. The Polo-like kinase 2 (PLK2) is also important for accurate spindle orientation in the mammary gland, although whether this is through the same route as Aurora A is not known[18]. Another crucial kinase is the Abelson murine leukemia viral oncogene homolog 1 (ABL1), which phosphorylates NuMA on a distinct residue than CDK1 or Aurora A, thus ensuring the maintenance of cortical NuMA during metaphase[19].

Cell-intrinsic components besides ternary complex members are also important for proper spindle positioning in human cells. This is the case of the acto-myosin network, which ensures cortical rigidity and thereby prevents the plasma membrane from being pulled inwards by forces exerted by astral microtubules[20]. Furthermore, the Ste20-like kinase (SLK) activates ezrin/radixin/moesin (ERM) proteins, which link cortical actin to the plasma membrane, with the loss of SLK or ERM proteins decreasing the amount of LGN and NuMA at the metaphase cortex without influencing Gαi[21]. Moreover, the unconventional myosin Myo10, as well as the mitotic interactor and substrate of Plk1 (MISP) protein, is important for proper spindle positioning also through their ability to couple cortical actin with astral microtubules[22,23]. In addition, phosphorylation of the myosin regulatory light chain by AMP-activated protein kinase (AMPK) modulates the association between astral microtubules and the cortical acto-myosin network[24]. Furthermore, the tumor suppressor liver kinase B1 (LKB1) is needed for proper AMPK distribution and, potentially thereby, accurate spindle positioning[25].

Importantly, spindle positioning is also regulated by cell extrinsic cues[26] (reviewed in ref. [5]). Thus, in non-polarized systems such as human HeLa cells, the spindle aligns with respect to the substratum via integrin-mediated cell-substrate adhesion. Besides integrin-linked kinase, such alignment requires astral microtubules, the microtubule end-binding protein EB1, cortical dynein, the actin cytoskeleton, and the motor protein myosin X[27–29]. Here, cortical actin is thought to link substrate-derived cues to spindle positioning by connecting retraction fibers with astral microtubules[30].

Despite significant knowledge regarding the mechanisms governing spindle positioning in human cells, it is likely that a more comprehensive understanding could be achieved through the discovery of further components critical for this process. Accordingly, a live imaging RNA interference (RNAi)-based screen of 107 proteins associated with LGN, Gαi, and dynein/dynactin complexes identified a novel role for the actin capping binding protein Zβ in spindle positioning, a role exerted through an impact on cortical dynein and microtubule dynamics[31]. Moreover, a small interfering RNA (siRNA)-based screen of 719 kinases and kinase-related components has been performed on fixed cells, thereby capturing endpoint phenotypes[19]. Although this screen uncovered notably the role of ABL1 in spindle positioning, such an endpoint assay might have missed some key components.

Here, therefore, we design and execute a large-scale unbiased live imaging functional genomic siRNA-based screen for components mediating spindle positioning in cultured human cells, screening a set of 1280 genes that comprises notably all kinases and phosphatases. We thus identify 16 components whose inactivation dramatically perturbs spindle positioning, including tyrosine receptor kinase 3 (TYRO3) and cyclin G-associated kinase (GAK). Follow-up experiments establish that TYRO3 functions by restricting the extent of cortical NuMA and dynein during metaphase, whereas GAK acts by ensuring the presence of robust astral microtubules during mitosis.

## Results

**Assay development.** We set out to design an siRNA-based live imaging functional genomic screen to identify novel modulators of spindle positioning in human cells. To visualize the position of the metaphase plate in time-lapse recordings, we utilized HeLa cells carrying an integrated plasmid expressing mCherry::H2B (Fig. 1a). Moreover, we took advantage of the ability of the substratum to dictate spindle positioning, such that the metaphase plate of cells grown on L-shaped fibronectin-coated micropatterns is typically positioned with a 45° angle to the arms of the L (Fig. 1a)[26].

As summarized in Fig. 1b and described in more detail in the Methods section, we developed a robust screening pipeline to identify spindle positioning phenotypes. In brief, HeLa mCherry::H2B cells were reverse transfected in 96-well plates containing siRNAs directed against genes to be tested, as well as negative controls (ctrl) and positive controls (LGN, which impairs but does not abolish spindle positioning)[2] (Fig. 1b). After incubation for 48h, cells were transferred to 96-well imaging plates containing L-shaped micropatterns, followed by the imaging of two visual fields per well once every 8 min during 24h (Supplementary Fig. 1a, b). To determine spindle position from the resulting recordings, we used the ImageJ-based pipeline TRACMIT to extract the angle of the metaphase plate with respect to the arms of the L-shape just before anaphase[32] (Fig. 1c, d).

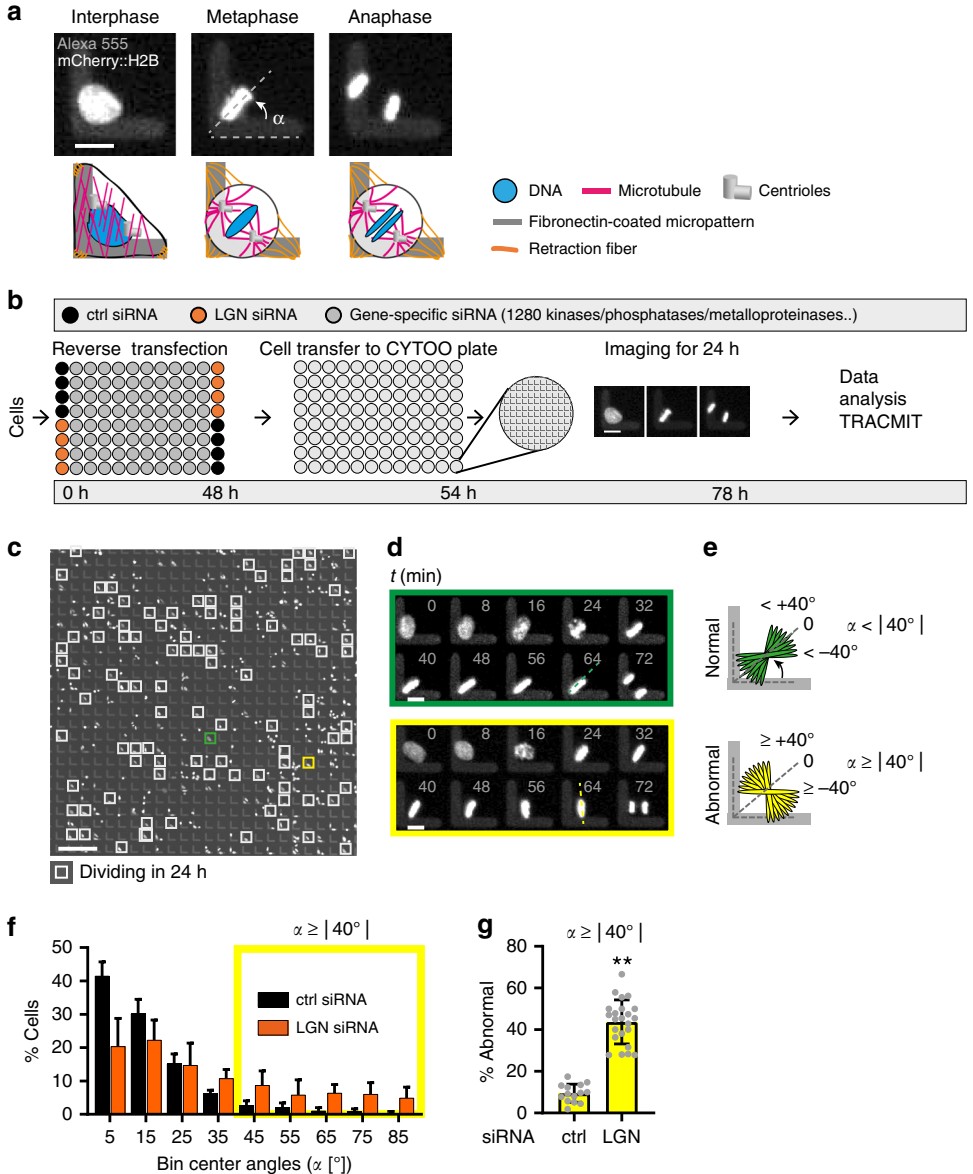

**Fig. 1** Live imaging functional genomic assay for spindle positioning in human cells. **a** HeLa cell expressing mCherry::H2B grown on L-shaped micropattern coated with fibronectin-Alexa 555. Top: raw time-lapse microscopy data; bottom: corresponding schematics. Upon cell rounding in mitosis, the metaphase plate is positioned with an angle $\alpha$ of ~45° (dashed line) with respect to the arms of the L. Scale bar: 10 μm. **b** Screening pipeline. Time in hours is indicated underneath. Cells (HeLa, mCherry::H2B) are seeded in small interfering RNA (siRNA)-containing 96-well plates. After incubation for 48 h, cells are transferred to 96-well plates containing L-shaped micropatterns and imaged for 24 h with a frame rate of 8 min (see **d**). Data analysis is performed using the ImageJ-based analysis pipeline TRACMIT. Scale bar: 10 μm. **c** Example of visual field from time-lapse microscopy (see **b**). Gray boxes mark micropatterns containing single cells that have divided within the 24 h imaging period. Green and yellow boxes indicate cells enlarged in **d**. Scale bar: 150 μm. **d** Green rectangle: cell dividing as expected (normal), with a metaphase angle close to the 0 reference position; yellow rectangle: cell deviating ≥40° from that position (abnormal spindle positioning). Time is indicated in min. Scale bar: 10 μm. **e** Schematic representations corresponding to **d**. Upper panel: "normal" spindle angles (green, $\alpha$ < 40° from 0 position); lower panel: "abnormal" spindle angles (yellow, $\alpha$ ≥ 40° from 0 position). **f** Frequency distributions of metaphase angles of control (ctrl, black) and LGN (orange) siRNA-treated cells from three pilot 96-well plates containing L-shaped micropatterns. Most control cells exhibit little deviation from the 0 reference position (bins 0–35°). Shown are mean values ± SDs, $p$ < 0.01, Mann–Whitney test was used to compare distributions between two groups, n: ctrl siRNA: 354, LGN siRNA: 334. Yellow box indicates abnormally positioned metaphase plates according to the definition in **e**. **g** Screen readout: mean percentage of cells with abnormal spindle positioning (i.e. $\alpha$ ≥ 40°) per well after treatment with control siRNA and LGN siRNA in the three pilot 96-well plates, ±SDs; **$p$ < 10$^{-4}$, Mann–Whitney $U$ test, n: ctrl siRNA: 354, LGN siRNA: 334

Three 96-well plates containing L-shaped micropatterns were used to test if metaphase angles in cells treated with ctrl and LGN siRNAs could be adequately discriminated. We refer to the position where the metaphase plate is at 45° from either arm of the L-shape as the normal position, and set it to 0 hereafter (Fig. 1e). Cells with perturbed spindle positioning are expected to exhibit metaphase plate angles away from this position. Analyzing the outcome of the three test plates using genetic programming[33] allowed us to establish that a metaphase plate angle ≥40° from the 0 position was the best discriminator between positive and negative controls (Supplementary Fig. 1c–f). Furthermore, the best robust strictly standardized mean difference (rSSMD), which

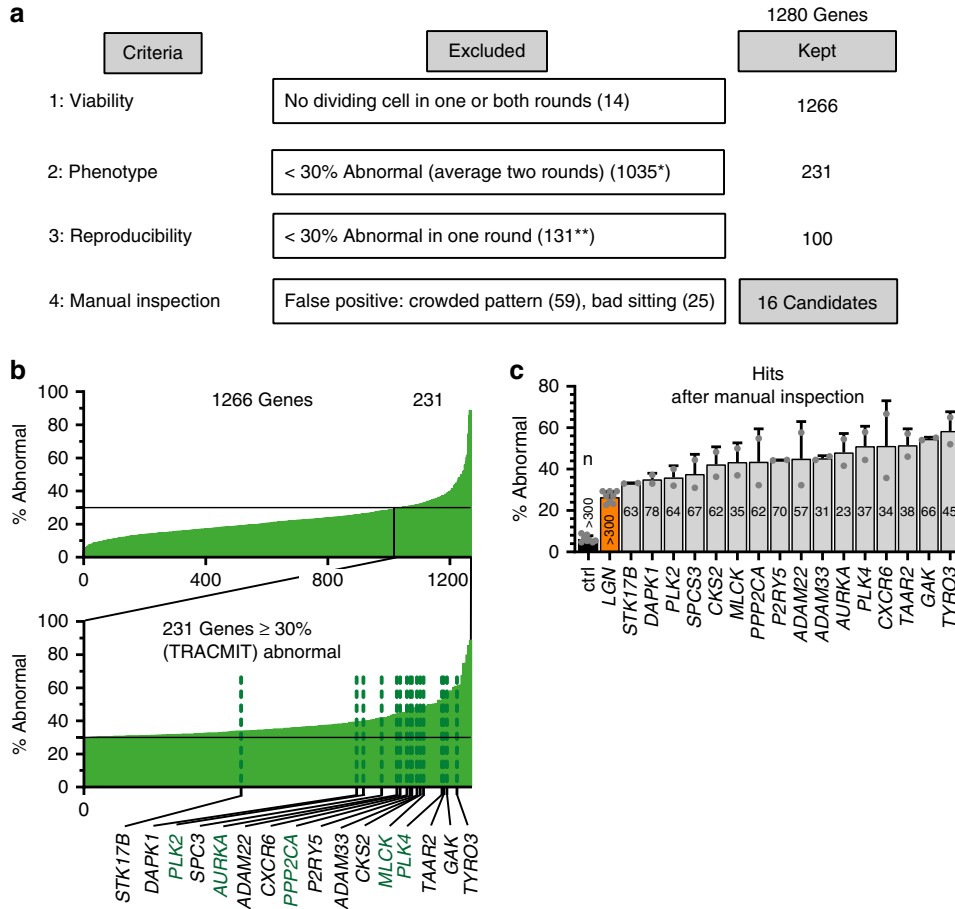

**Fig. 2** Identification of 16 spindle positioning candidate genes. **a** Schematic representation of step-wise hit selection procedure following two rounds of screening a small interfering RNA (siRNA) collection targeting 1280 kinases, phosphatases, metalloproteases, G protein-coupled receptors and associated proteins. Left column: four criteria that were evaluated successively; middle column: exclusion criteria and number of excluded genes at each step; right column: number of genes kept after each step. *Including one gene with ≤10 cells to analyze and one gene with only two cells to analyze in one round, and a manually confirmed phenotype in the other round (PPP1CB). **Including eight genes with ≤10 cells to analyze and two genes with >10 cells in one round, and a manually confirmed phenotype in the other round, but with <20 cells to analyze (*EFNB3, PRKACG*, see also Supplementary Data 1). **b** Graphical representation of screen outcome, with magnification of the 231 candidates exhibiting a phenotype in >30% of all cells analyzed in both rounds. The 16 candidates finally retained are singled out. Green lettering: candidates anticipated from prior work. **c** Control (ctrl) siRNA, as well as siRNAs against LGN or one of the 16 candidates remaining after manual inspection of all movies corresponding to the 106 candidates obtained in step 3 of the hit selection procedure (see **a**). Shown are mean values of the percentage of abnormal cells per well in the two rounds of screening combined ±SDs and the total number of dividing cells analyzed in each case

discriminates positive and negative controls based on differences in their medians as well as in median absolute deviation[34,35], were obtained using the ≥40° angle offset criterion (Supplementary Fig. 1g). Therefore, the percentage of cells per well exhibiting a metaphase plate angle ≥40° was used as the primary screen readout (% abnormal; Fig. 1e, f, yellow area). For the three test 96-well plates, this percentage was on average ~10% in the negative control and ~40% in cells treated with LGN siRNAs (Fig. 1g). Overall, we conclude that we have developed a 96-well plate based live imaging screening pipeline for spindle positioning defects in human cells.

**Live imaging functional genomic screen for spindle positioning defects in human cells.** We used this screening pipeline to probe an siRNA library with four different siRNAs per gene mixed in one well, representing 1280 kinases, phosphatases, metalloproteases, some G-protein coupled receptors, and related proteins (see Methods section). We screened the library twice, analyzing on average 120 cells per siRNA condition (first and

second round averages: 64 and 56 cells, respectively; Supplementary Fig. 2a; Methods). Overall, we imaged ~178,000 mitotic cells whose metaphase spindle position was determined using TRACMIT. The entire time-lapse microscopy data set is available in the Image Data Resource (IDR) [https://idr.openmicroscopy.org/] under accession number idr0061. For quality control, we determined the rSSMD value of each plate and repeated those in which this value was <2 (Supplementary Fig. 2c; Methods)[35].

Candidate spindle positioning genes were identified using a step-wise procedure (Fig. 2a, Supplementary Data 1, Methods). In a first step, 14 genes were excluded because no cells were present in at least one round of screening, leaving 1266 genes for further analysis. Second, we set an empiric cutoff at ≥30% of cells having to exhibit abnormal spindle positioning when combining the numbers from both screening rounds. We choose ≥30% to select candidates with strong spindle positioning phenotypes; LGN siRNA has a weaker phenotype, with ~26% of cells (±10%, n = 256 wells) exhibiting abnormal spindle positioning[2] (Supplementary Fig. 2b). This second filter left 231 genes to consider further (Fig. 2b). Third, we excluded 131 of these genes because their depletion led to

abnormal spindle positioning in ≥30% of cells in only one of the two screening rounds. Fourth, movies corresponding to the two rounds for the remaining 100 genes were inspected manually. This led to the exclusion of 84 genes that were false positives owing to single cells being poorly located on the micropatterns, or else to more than one cell being present on one micropattern[32]. Overall, the above step-wise procedure led to the identification of 16 candidates spindle positioning genes (Fig. 2a–c). Representative movie sequences during mitosis following depletion of these 16 components are reported in Supplementary Fig. 3.

The 16 candidate genes comprised 5 genes that were described previously as having an impact on spindle positioning in mammalian cells. These are the kinases Aurora A[15,16] and PLK2[18], as well as the phosphatase PPP2CA[12]. As anticipated, the screen also identified PLK4, whose depletion causes impaired centriole duplication, thereby resulting in spindle pole asymmetry and aberrant spindle positioning[36]. Moreover, we identified the myosin light chain kinase, which is required for positioning the meiotic spindle in the mouse oocyte[37]. In addition to these five genes, another five that might have been expected to be identified based on previous work did not meet the stringent criteria of the above step-wise procedure (Supplementary Table 1). Therefore, we identified 5/10 of the genes that were expected from prior work to be associated with a spindle positioning phenotype. By extension, we estimate the false-negative discovery rate of the present screen to be ~50%. Overall, we conclude that our live imaging screening pipeline has identified 16 candidate modulators of spindle positioning in human cells.

**Further validation of TYRO3 and GAK**. For further validation and analysis, we selected TYRO3 and cyclin GAK, the two strongest hits among the 16 candidates, for which a role in spindle positioning was not reported previously.

We ascertained that TYRO3 and GAK siRNA-mediated depletion spindle positioning phenotypes could be observed also by assaying fixed HeLa, hTERT-RPE-1, and U2OS cells plated on uniform fibronectin-containing coverslips[38] (Fig. 3a–d). The spindle positioning phenotypes in HeLa cells upon TYRO3 or GAK depletion were not due merely to a prolonged metaphase, since there was no correlation between metaphase duration and strength of the spindle positioning phenotype in TYRO3- or GAK-depleted cells (Supplementary Fig. 2d). Moreover, given that TYRO3 or GAK depletion can lead to lagging chromosomes (Supplementary Fig. 2e)[39], which itself can cause spindle positioning phenotypes[40], we excluded such cells from our analysis. Furthermore, we found that spindle positioning phenotypes upon TYRO3 or GAK depletion were not related to altered cell dimensions (Supplementary Fig. 4a–c), centriole numbers (Supplementary Fig. 4d), or differences in cortical actin (Supplementary Fig. 4e–h).

We sought to confirm the TYRO3 or GAK depletion spindle positioning phenotypes with different means. We first used two other siRNAs (siRNA A and B) for each gene, which severely depleted the corresponding components, as evidenced by reverse transcription-polymerase chain reaction and Western blot analyses (Supplementary Fig. 4i, k, l). These siRNAs were utilized to assay spindle positioning by live imaging of HeLa cells expressing mCherry::H2B and plated on L-shaped micropatterns, as well as by analyzing fixed HeLa cells plated on uniform fibronectin-containing coverslips. Both sets of experiments confirmed the pronounced spindle positioning phenotype upon TYRO3 or GAK depletion (Fig. 3e, Supplementary Fig. 4m). Moreover, we sought to generate CRISPR/Cas9 knockout (KO) cell lines for both genes (Supplementary Fig. 4j, k, l). The two TYRO3 and the two GAK CRISPR/Cas9 cell lines tested also displayed clear spindle positioning defects (Fig. 3f), although we

noted that the phenotype became weaker with increasing cell passage number, indicative of potential compensatory mechanisms. The spindle positioning phenotype of TYRO3 CRISPR/Cas9 cells and of cells depleted of GAK using siRNAs was rescued by providing plasmids encoding the corresponding proteins (Fig. 3g, see also Fig. 6a–e). Moreover, we found that adding TYRO3-siRNA to TYRO3 CRISPR/Cas9 cells did not worsen the phenotype, further indicating that the consequences of TYRO3 depletion by siRNAs are not due to off-target effects (Fig. 3g). Overall, we conclude that both TYRO3 and GAK are bona fide spindle positioning modulators.

**TYRO3 restricts metaphase cortical NuMA/dynein and PIP$_2$ accumulation**. TYRO3 is a single-pass type 1 transmembrane protein that belongs to the TAM (TYRO3, AXL, MERK) transmembrane receptor kinase family, whose members transduce signals from the extracellular matrix into the cytoplasm upon binding of dedicated ligands[41]. Although TYRO3 functions at the plasma membrane in other settings, the localization of the protein during mitosis has not been previously reported. Using immunofluorescence analysis of mitotic HeLa cells stained with TYRO3 antibodies, we found that the plasma membrane signal was highest during metaphase (Fig. 4a–c), which is notable given the spindle positioning phenotype. This signal is specific, as evidenced by its absence in cells treated with siRNAs against TYRO3 and in the TYRO3 CRISPR/Cas9 cell lines (Fig. 4d, e). By contrast, the centrosomal signal also detected by these antibodies in control cells does not appear to be specific, since it remains present to a substantial extent in TYRO3 CRISPR/Cas9 cells (Fig. 4d) or upon TYRO3 siRNA (Fig. 4e, Supplementary Fig. 4n).

In order to further characterize the TYRO3 depletion spindle positioning phenotype, we conducted spinning disk confocal microscopy of HeLa Kyoto cells seeded in fibronectin-coated dishes and expressing EGFP::α-tubulin and mCherry::H2B[42]. In contrast to control cells, in which the spindle moved only minimally during metaphase (Fig. 4f; Supplementary Movie 1), we found that cells depleted of TYRO3 using siRNAs underwent extensive metaphase spindle oscillations (Fig. 4g; Supplementary Movie 2). This phenotype resembles conditions in which excess cortical force generators are present, as upon targeting of excess Gαi to the plasma membrane[43]. Therefore, we investigated whether excess NuMA and dynein are present at the cortex of cells depleted of TYRO3. Using immunofluorescence analysis with antibodies against NuMA and the dynein-associated component p150[Glued], we found this to be the case indeed (Fig. 4h–j, Supplementary Fig. 4o, p), whereas NuMA protein levels in the cell and at spindle poles remained unaltered (Supplementary Fig. 5a–c). Moreover, we imaged HeLa cells expressing the dynein heavy chain fused to green fluorescent protein (GFP) as they transited through mitosis. In control conditions, cortical DHC::GFP was minimal during metaphase, but then increased in anaphase (Fig. 4k–o; Supplementary Movie 3), as previously reported[6]. In stark contrast, we found that upon TYRO3 depletion, cortical DHC::GFP was high already during metaphase and did not increase further in anaphase (Fig. 4m–o; Supplementary Movie 4). Together, these observations indicate that TYRO3 depletion results in excess NuMA at the cell cortex of metaphase cells.

How does TYRO3 normally act to limit NuMA and dynein cortical distribution? We set out to address whether TYRO3 kinase activity is important. To this end, we treated HeLa cells plated on fibronectin-coated coverslips with the pan-TAM inhibitor BMS777607[44], which resulted in a clear spindle positioning phenotype (Supplementary Fig. 5d). Although BMS777607 also inhibits AXL and MERK[44], given that these

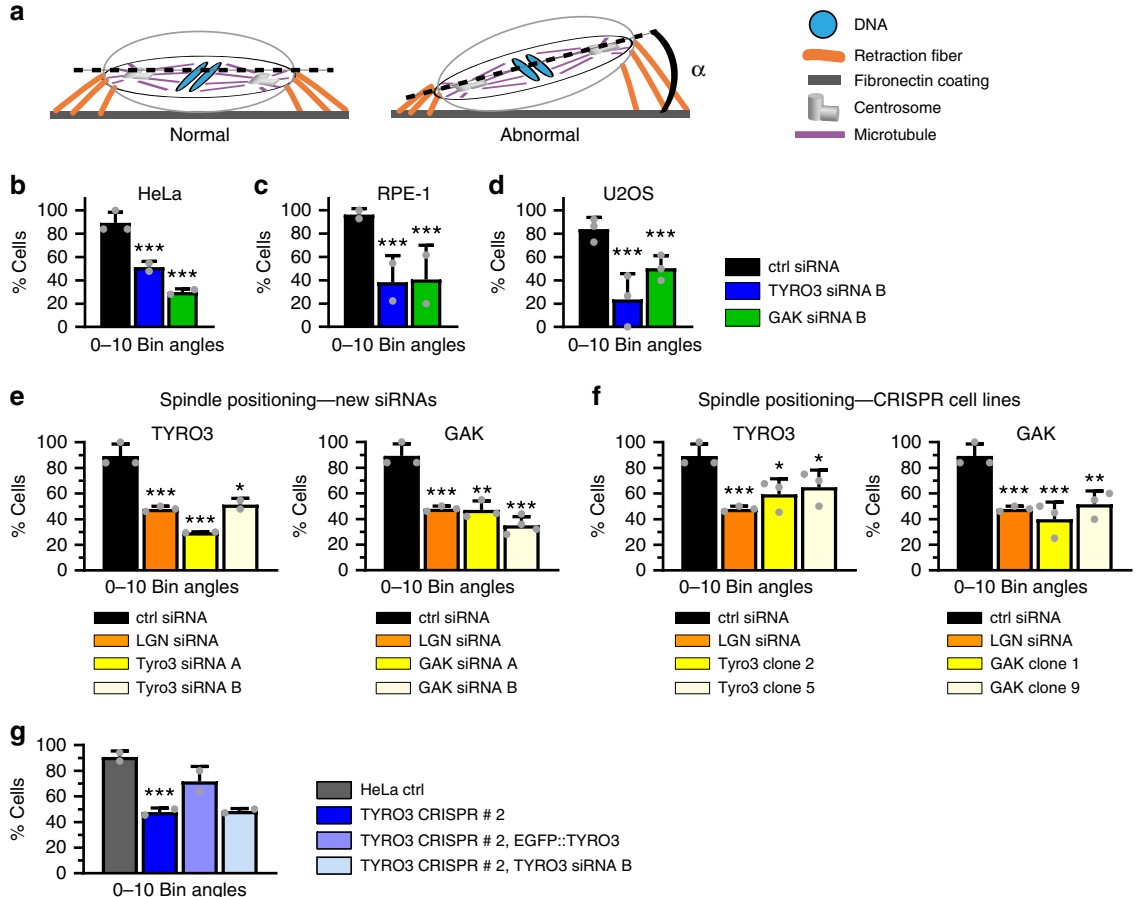

**Fig. 3** Validation of cyclin G-associated kinase (GAK) and tyrosine receptor kinase 3 (TYRO3) spindle positioning phenotypes. Shown are percentages of cells with spindle positioning angles ($\alpha$) between 0 and 10°. **a** Schematic representation of a control cell (left) or a cell with abnormal spindle positioning (right) seeded on a coverslip coated with uniform fibronectin as used in experiments reported in **b–g**. **b–d** Mean values (±SD) of spindle positioning angle $\alpha$ in HeLa (**b**), RPE-1 (**c**), or U2OS cells (**d**). Number of cells analyzed from two (**d**) or three (**b**, **c**) independent experiments: HeLa ctrl: 30, HeLa TYRO3: 37, HeLa GAK: 37, RPE-1 ctrl: 42, RPE-1 TYRO3: 39, RPE-1 GAK: 35, U2OS ctrl: 65, U2OS TYRO3: 52, and U2OS GAK: 47. Statistical analyses were conducted using Mann–Whitney $U$ test, ***$p < 0.001$. **e**, **f** Mean values (±SD) of spindle positioning angle $\alpha$ of HeLa cells treated using new small interfering RNAs (siRNAs) (**e**), as well as CRISPR/Cas9 cell lines (**f**). Number of cells analyzed from two independent experiments each, unless stated otherwise: ctrl: 84, LGN: 34, TYRO3 siRNA A: 30, TYRO3 siRNA B: 31, GAK siRNA A: 33, GAK siRNA B: 42, TYRO CRISPR/Cas9 # 2: 42 (three independent experiments), TYRO3 CRISPR/Cas9 # 5: 31, GAK CRISPR/Cas9 # 1: 40 (three independent experiments), GAK CRISPR/Cas9 # 9: 31. Mann–Whitney $U$ test was used to compare entire angle distributions between groups exhibiting non-Gaussian distribution, all ***$p < 0.0001$, ctrl vs. TYRO3 siRNA B: *$p = 0.043$, ctrl vs. GAK siRNA A: **$p = 0.0013$, ctrl vs. CRISPR/Cas9 TYRO3 # 2: *$p = 0.02$, ctrl vs. CRISPR/Cas9 TYRO3 # 5: *$p = 0.04$, ctrl vs. CRISPR/Cas9 GAK # 9: **$p = 0.002$. **g** Mean values (±SD) of spindle positioning angle $\alpha$ of HeLa ctrl cells, HeLa TYRO3 CRISPR/Cas9 # 2 cells, TYRO3 CRISPR/Cas9 # 2 cells transfected with EGFP::TYRO3, and TYRO3 CRISPR/Cas9 # 2 cells transfected with TYRO3 siRNA. Number of cells analyzed from two independent experiments each: HeLa ctrl: 33, HeLa TYRO3 CRISPR/Cas9 # 2: 35, TYRO3 CRISPR/Cas9 # 2 transfected with EGFP::TYRO3: 42, TYRO3 CRISPR/Cas9 transfected with TYRO3 siRNA: 32. Statistical analyses were conducted with Mann–Whitney $U$ test. ctrl vs. TYRO3 CRISPR/Cas9 # 2: ***$p < 0.001$: ctrl vs. TYRO3 CRISPR/Cas9 # 2 plus EGFP::TYRO3: $p = 0.1267$: TYRO3 CRISPR/Cas9 # 2 vs. TYRO3 CRISPR/Cas9 # 2 plus TYRO3 siRNA: $p = 0.5972$

two family members do not appear to be involved in spindle positioning, this result supports the notion that TYRO3 kinase activity is important for this process. Suggestively, NuMA associates with $PIP_2$ independently of LGN during anaphase[12,13], and TYRO3 signaling can activate phosophoinositide 3-kinase (PI3K), which transforms $PIP_2$ to $PIP_3$. Therefore, we hypothesized that TYRO3 depletion might impair PI3K and thereby result in elevated plasma membrane $PIP_2$, thus potentially explaining the spindle positioning phenotype. Compatible with this possibility, monitoring $PIP_2$ at the plasma membrane using GFP-C1-PLCδ[45] revealed a significant increase after TYRO3 depletion (Supplementary Fig. 5g, h). Moreover, blockade of PI3K with the small-molecule LY294002 mimics the TYRO3 siRNA phenotype (Supplementary Fig. 5e, f; Supplementary Movies 5, 6), as previously reported[12,38].

Taken together, our findings uncover that the presence of TYRO3 is crucial for limiting NuMA and dynein levels at the cell cortex during metaphase, potentially through a role in PI3K signaling.

**GAK acts with Clathrin to ensure robust astral microtubules and spindle positioning.** GAK protein localizes to the Golgi and the perinuclear region in interphase cells[46], as well as to centrosomes in mitosis[47]. Using immunofluorescence analysis of HeLa cells stained with GAK antibodies (Fig. 5a–b), as well as anti-myc immunofluorescence of cells carrying myc-tagged GAK (Supplementary Fig. 6a), we confirmed that the protein localizes to spindle poles in mitotic cells (Fig. 5a). The signal is specific, given its absence in cells treated with siRNA against GAK (Fig. 5b). We did not detect a centrosomal localization in interphase cells

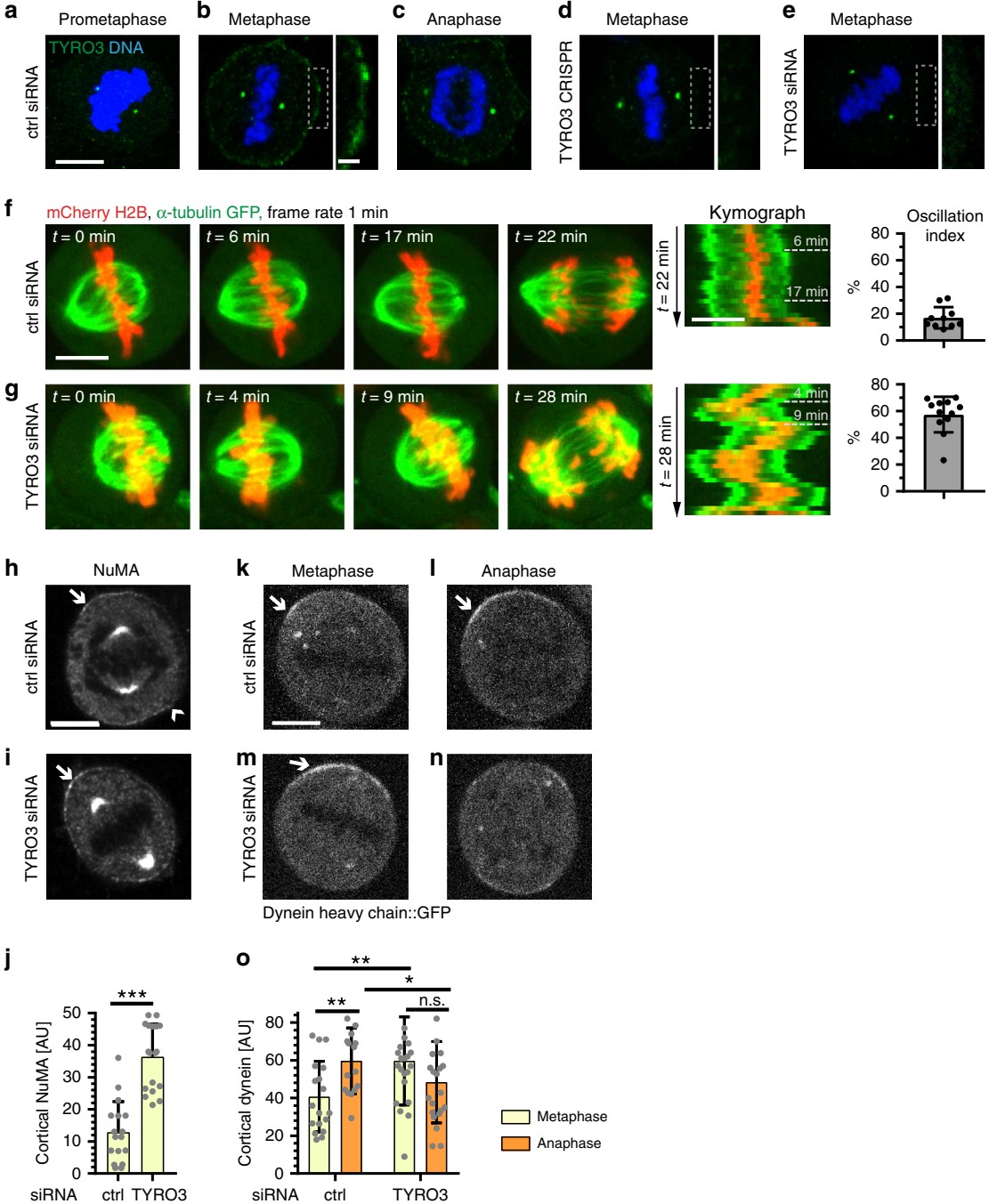

**Fig. 4** Tyrosine receptor kinase 3 (TYRO3) limits metaphase cortical NuMA/dynein accumulation. **a–e** Immunofluorescence of HeLa cells treated with ctrl small interfering RNAs (siRNAs) (**a**, prometaphase; **b**, metaphase; **c**, anaphase), TYRO3 CRISPR/Cas9 cells (**d**, metaphase), as well as TYRO3 siRNA (**e**, metaphase), all stained with TYRO3 antibodies (green). DNA in blue. Scale bar: 10 μm. The insets of the cortical regions boxed in **b**, **d**, and **e** are magnified on the right in each case. Scale bar: 2 μm. The two dots on each side of the metaphase plate correspond to unspecific centrosomal signal. **f**, **g** Stills (left) and corresponding kymographs (right, with position of stills indicated) from live imaging of HeLa expressing mCherry::H2B and GFP::α-tubulin after ctrl, $n = 10$ cells (**f**) or TYRO3 siRNA, $n = 11$ cells (**g**), from three independent experiments each, with corresponding mean oscillation indices values ± SD. Scale bars: 10 μm, ***$p = 6 \times 10^{-8}$, comparing ctrl and TYRO3 siRNA oscillation indices using unpaired two-tailed Student's $t$ test with Welch's correction. **h**, **i** Immunofluorescence analysis of metaphase HeLa cells treated with ctrl or TYRO3 siRNAs and stained with NuMA antibody. Arrows point to cortical signal. **j** Quantification of cortical signal corresponding to **h** and **i**. Shown are mean values (in arbitrary units, AU, ±SD) of background-corrected cortical intensities (Methods) from four independent experiments, in each case quantifying 3–5 cells per group, yielding a total of 17 cells for ctrl and 18 cells for TYRO3 siRNAs; ***$p = 2 \times 10^{-7}$, Mann–Whitney $U$ test. **k–n** Still images from live imaging of HeLa cells in either metaphase or anaphase, as indicated, carrying dynein heavy chain::GFP and treated with ctrl (**k**, **l**) or TYRO3 (**m**, **n**) siRNAs. **o** Quantification of cortical green fluorescent protein (GFP) signal from **n** to **q** in metaphase and anaphase. Shown are mean values (±S.D.) of $n = 18$ cells for control metaphase and anaphase, $n = 23$ cells for TYRO3 metaphase, and $n = 22$ cells for TYRO3 anaphase from three independent experiments each. Statistical analyses were conducted with unpaired two-tailed Student's $t$-test with Welch's correction; ctrl meta vs. ana: **$p = 0.003$, ctrl meta vs. TYRO3 meta: **$p = 0.0065$, ctrl ana vs. TYRO3 ana: *$p = 0.0446$, TYRO3 meta vs. TYRO3 ana: $p = 0.09$. n.s. not significant

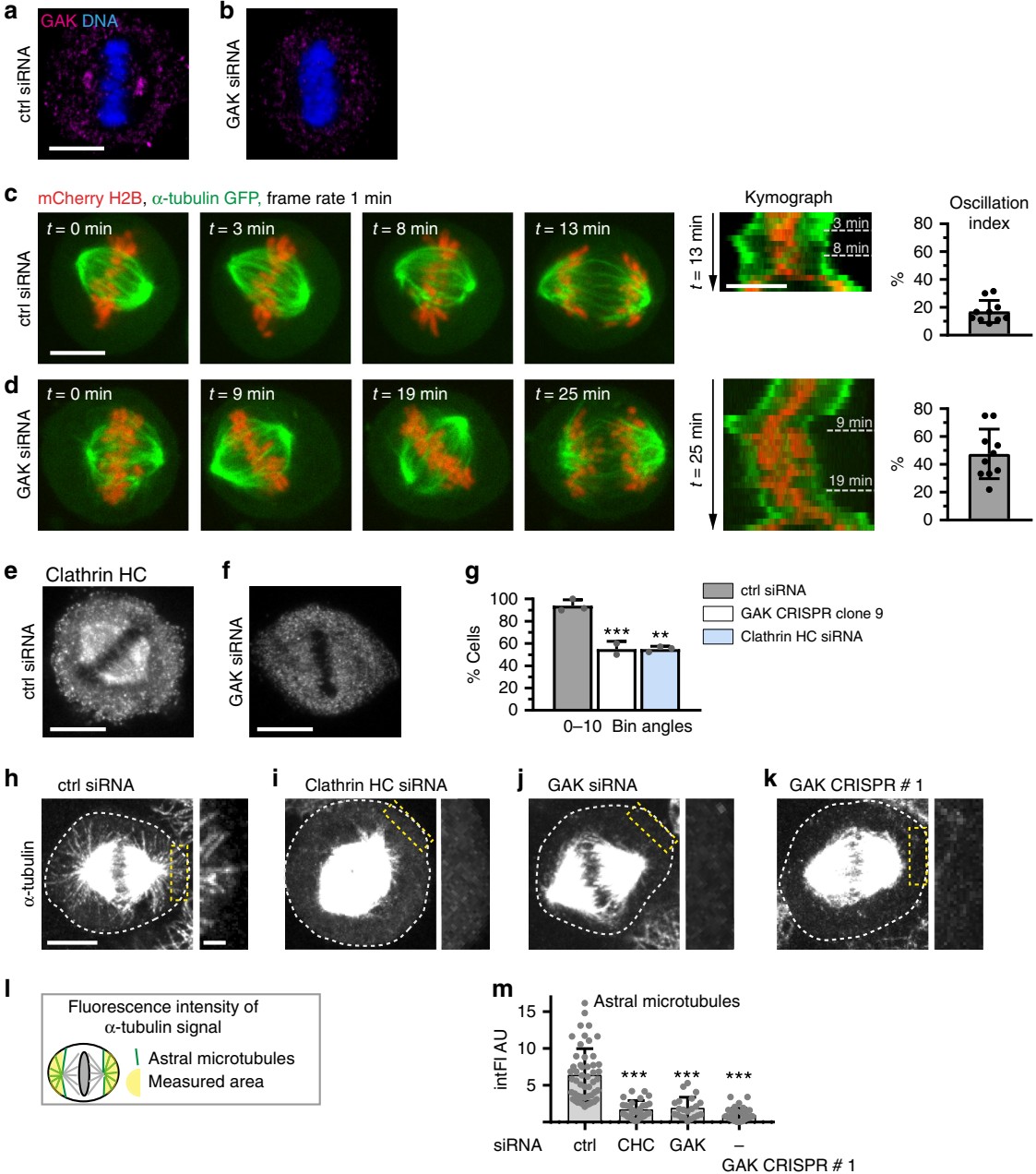

**Fig. 5** Cyclin G-associated kinase (GAK) promotes the presence of astral microtubules during spindle positioning. **a**, **b** Immunofluorescence of metaphase HeLa cells treated as indicated and stained with GAK antibodies (red); DNA in blue. Scale bar: 10 μm, throughout the figure, except where noted. **c**, **d** Stills (left) and corresponding kymographs (right, with positions of stills) from live imaging of HeLa cells expressing mCherry::H2B and GFP::α-tubulin after ctrl, $n = 10$ cells (**c**) or GAK, $n = 10$ cells (**d**) small interfering RNA (siRNA) treatment, three independent experiments each, with corresponding mean oscillation indices values ± SD, $p = 0.0003$ comparing ctrl and GAK siRNA, unpaired two-tailed Student's $t$ test with Welch's correction. **e**, **f** Immunofluorescence of metaphase HeLa cells treated as indicated and stained with Clathrin heavy chain antibodies (Clathrin HC). **g** Spindle positioning in cells grown on fibronectin-coated coverslips and treated with ctrl ($n = 34$ cells, three independent experiments) or Clathrin heavy chain siRNA ($n = 37$ cells, three independent experiments); GAK CRISPR/Cas9 clone 9 is also shown ($n = 18$ cells, two independent experiments). Shown are mean percentage of cells ±SD, aligning their mitotic spindles at 0–10° with respect to the surface; raw angle distributions were compared using Mann–Whitney $U$ test, ***$p = 0.0004$: ctrl vs. GAK CRISPR, **$p = 0.0066$: ctrl vs. Clathrin siRNA. Note that only cells with robust central spindle microtubules were analyzed to avoid potential phenotypes stemming from spindle assembly defects. **h–k** Immunofluorescence of metaphase HeLa cells treated with indicated siRNAs and stained with α-tubulin antibodies. Imaging was conducted so as to enable visualization of astral microtubules, if present. Insets show magnifications of indicated cortical regions; dashed lines: cell boundaries. Scale bar in inset: 2 μm. **l**, **m**: Measurements of astral microtubule α-tubulin fluorescence corresponding to (**h–k**) in the area indicated in **l**. Shown are averages of two poles per cell in z-projections of relevant optical planes (in arbitrary units, AU, ±SD). The number of cells analyzed from two independent experiments each were: ctrl siRNA: 52, Clathrin heavy chain siRNA (CHC): 29, GAK siRNA: 21, GAK CRISPR cell line: 37, ctrl vs. CHC: ***$p = 3 \times 10^{-12}$, ctrl vs. GAK: ***$p = 1 \times 10^{-10}$, ctrl vs. GAK CRISPR # 2: ***$p = 8 \times 10^{-15}$, unpaired two-tailed Student's $t$ test with Welch correction

(Supplementary Fig. 6a), suggestive of a transient association of GAK with spindle poles.

In order to further characterize the GAK depletion spindle positioning phenotype, we conducted spinning disk confocal microscopy as described above for TYRO3. We found that cells depleted of GAK underwent slow drifting spindle movements, with metaphase plates turning around all axes in 3D (Fig. 5c, d; Supplementary Movies 7 and 8). We found no striking difference upon GAK depletion in the distribution of NuMA or DHC::GFP when compared to control cells (Supplementary Fig. 6b, c).

How, then, does GAK depletion lead to a spindle positioning phenotype? We explored whether Clathrin may be involved, since GAK is responsible for uncoating Clathrin-coated vesicles[46] and for the presence of Clathrin on the central spindle during mitosis[39]. We confirmed the requirement of GAK for such central spindle distribution of Clathrin using both siRNAs and CRISPR/Cas9 cells (Fig. 5e, f, Supplementary Fig. 6d, e)[39]. Importantly, we found that siRNA-mediated depletion of the Clathrin heavy chain (CHC) also leads to a spindle positioning phenotype (Fig. 5g). Because GAK depletion is known to dampen microtubule polymerization stemming from chromosomes[39] and given the spindle positioning phenotype uncovered here, we analyzed astral microtubules in cells depleted of GAK or the CHC. Cells with severe GAK depletion exhibited an apparent reduction of central

spindle microtubules, as previously reported[39] (Supplementary Fig. 6f). Although this can lead to lagging chromosomes[39] (Supplementary Fig. 2e), we restricted analysis of spindle positioning to cells with correctly aligned chromosomes to avoid potentially confounding effects. Interestingly, we discovered that there was a striking reduction of astral microtubules in cells depleted of GAK or of the CHC, including in mildly affected cells in which central spindle microtubules appeared normal (Fig. 5h–m). We conclude that GAK and Clathrin are essential for the presence of astral microtubules during mitosis, and likely thereby for proper spindle positioning.

We next sought to identify domains within the GAK protein responsible for proper spindle positioning. GAK contains a kinase domain, as well as a Clathrin-binding domain and a J domain responsible for uncoating of Clathrin-coated vesicles[48]. We depleted GAK using siRNAs targeting the GAK 3′-untranslated region (UTR) and expressed either full-length GAK or various truncation or mutant constructs, scoring the resulting spindle positioning angles (Fig. 6a–e). This analysis revealed that both wild-type and kinase-dead full-length constructs rescued the spindle positioning phenotype (Figs. 6a, b). Moreover, the kinase domain alone exhibited partial rescue (Fig. 6c), as did a construct lacking the J domain (Fig. 6d). Together, these findings indicate that robust function of GAK in spindle positioning requires the J

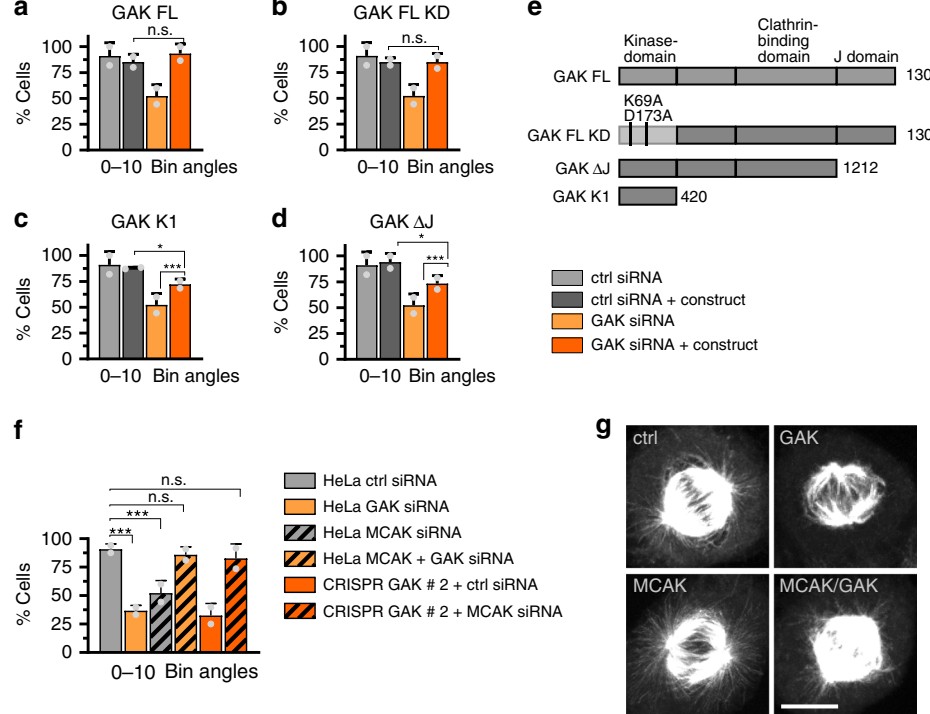

**Fig. 6** Function of cyclin G-associated kinase (GAK) in spindle positioning requires J domain. **a–f** Spindle positioning phenotype of cells grown on fibronectin-coated coverslips. Shown are mean percentages of cells ±SD with mitotic spindles aligned at 0–10° with respect to the surface. **a–e** Cells treated with ctrl or GAK small interfering RNA (siRNA), as indicated, and expressing one of the GAK constructs shown schematically in **e**. GAK FL: GAK full-length, GAK FL KD: kinase dead version, GAK ΔJ: lacking the J domain, GAK K1: kinase domain only; metaphase angles are reported. Numbers of cells analyzed in two independent experiments each: control siRNA (ctrl): 22, GAK siRNA (GAK): 38, ctrl siRNA + GAK FL: 31, GAK siRNA GAK FL: 21, ctrl siRNA + GAK KD: 44, GAK siRNA + GAK KD: 44, ctrl siRNA + GAK ΔJ: 64, GAK siRNA + GAK ΔJ: 67, ctrl siRNA + GAK KD: 57, GAK siRNA + GAK KD: 75. Angles were compared between the group ctrl siRNA + GAK construct (dark gray) and the group GAK siRNA + GAK construct (dark orange), as well as GAK siRNA vs. GAK siRNA plus construct (**p**, **q**) using a Mann–Whitney U test. $p = 0.60$ (**a**), $p = 0.77$ (**b**), *$p = 0.0143$ (**d**), *$p = 0.025$ (**c**), ***$p = 1.6 \times 10^{-5}$ (**d**), ***$p = 4.5 \times 10^{-5}$ (**c**), n.s. = non significant. **f** Cells treated with ctrl (n = 23 cells), MCAK siRNA (n = 28 cells), GAK siRNA (n = 22 cells), MCAK siRNA plus GAK siRNA (n = 32 cells), GAK CRISPR/Cas9 # 1 plus ctrl siRNA (n = 31 cells) or GAK CRISPR/Cas9 # 1 plus MCAK siRNA (n = 31 cells). Results from two biological replicates (mean ± SD); angles distributions were compared using Mann–Whitney U test, ***$p = 0.0008$: ctrl vs. GAK siRNA, $p = 0.0002$: ctrl vs. MCAK siRNA, $p = 0.5850$: ctrl vs. GAK plus MCAK siRNA, $p = 0.0699$: ctrl vs. GAK CRISPR/CAS9 # 1 plus MCAK siRNA. **g** Immunofluorescence of cells treated with ctrl siRNA, GAK siRNA, MCAK siRNA or both, as indicated, and stained with antibodies against α-tubulin. Scale bar: 10 µm

domain, possibly reflecting an interaction with Clathrin, with an additional contribution from the kinase domain.

Clathrin is needed for the binding to spindle microtubules of GTSE1, an EB1-dependent plus-end tracking protein that can inhibit the microtubule depolymerase MCAK[49]. The absence of Clathrin results in failure of GTSE1 recruitment to spindle microtubules and to consequent rapid MCAK-mediated microtubule depolymerization[49]. Therefore, we set out to test the possibility that the phenotype upon GAK depletion might result from inappropriate MCAK activity following loss of Clathrin from the mitotic spindle. Importantly, we found that depleting MCAK in addition to targeting GAK either by siRNAs or using CRISPR/Cas9 rescued the spindle positioning phenotype incurred upon depletion of GAK alone (Fig. 6f, g). We conclude that GAK, potentially through Clathrin and GTSE1, negatively regulates MCAK and thus ensures a robust astral microtubule network and proper spindle positioning.

## Discussion

Highly dynamic cellular processes such as mitosis can be best studied with live imaging. Here, we report the first large-scale live imaging screen of spindle positioning in human cells, using stereotyped micropatterns to test the contribution of the compendium of kinases, phosphatases, and metalloproteases, as well as some G protein-coupled receptors and associated proteins of the proteome.

Using L-shaped micropatterns as a substratum results in the important advantage of stereotyping metaphase spindle positions. Obviously, since proper attachment of cells on the micropatterns is essential in this setting, genes with a dual function in cell adhesion and spindle positioning would have been missed. The same holds for genes with any other function that would have precluded scoring spindle positioning, such as arrest earlier in the cell cycle. Regardless of these limitations, the present extensive data set of spindle positioning phenotypes available in Supplementary Data 1 is anticipated to serve as an important resource for further investigation of this critical cellular process. Using this resource, one should recall that the threshold applied here to deem a gene a candidate spindle positioning component is based on LGN, the phenotypic severity of which can vary between experimental systems[3,8,21,31]. Therefore, weak or partially redundant regulators of spindle positioning are expected not to have been retained amongst the final 16 strong candidates in this work, but can be analyzed by mining Supplementary Data 1, as well as the entire time-lapse microscopy data set available in IDR. We note also that the results of the present study exhibit little overlap with those from a screen analyzing fixed cells on uniform fibronectin-coated 96-well plates[19]. The numerous technical differences between the two screens likely explain such lack of substantial overlap.

Developing and executing the live imaging screening pipeline enabled us to discover 16 robust candidate genes modulating spindle positioning in human cells, including 11 that had not been previously associated with such a function. We verified the phenotype using different siRNAs and CRISPR/Cas9-mediated impairment for the two strongest hits, TYRO3 and GAK. By extension, together with the finding of the five candidates previously known to exhibit a spindle positioning phenotype, we anticipate that most novel genes identified in the present screen represent bona fide components regulating spindle positioning in human cells.

The candidate with the strongest spindle positioning phenotype is TYRO3, a protein with many previously reported functions, including in controlling cell survival and proliferation, spermatogenesis, immunoregulation, and phagocytosis[50].

Compatible with our findings, depletion of TYRO3 exhibited significant alteration of spindle positioning in a screen using fixed cells on uniform fibronectin[19], but the $p$ value did not meet the empirical cutoff set in that study. Like the other TAM receptors, TYRO3 is absent from invertebrates[51], such that the spindle positioning phenotype uncovered here appears restricted to vertebrate systems. Moreover, TAM receptors are not known to function during embryogenesis[52], so that the regulation of spindle positioning uncovered here might apply strictly post-embryonically in vivo. In non-neural cells, TYRO3 substrates include the src kinase family members src[53] and FYN[54] and the Ran-binding protein RanBPM[55], as well as the regulatory subunit of PI3K p85[56]. Depletion of src leads to impaired microtubule outgrowth from centrosomes during prometaphase[57], whereas FYN has not been shown to be involved in spindle positioning thus far[57]. We note also that src or FYN depletion did not perturb spindle positioning in the present screen and that microtubule asters were not altered following TYRO3 depletion (Supplementary Data 1 and Supplementary Fig. 6f). Therefore, the function of TYRO3 in spindle positioning in human cells might not be exerted through src or FYN. While all three TAM receptors are expressed in HeLa cells[58], only TYRO3 was found to be a screen hit. Depletion of AXL indeed did not yield a spindle positioning phenotype in either screening round (23.6% phenotype in first round and 16% in second round). By contrast, between the two rounds of screening, 28% (22% and 34%) of cells depleted of MERTK exhibited a spindle positioning phenotype, thus just narrowly missing our stringent cutoff. Although many functions of TAMs are overlapping[58], single TAMs also have unique functions in some cases, as exemplified by the role of MERTK in phagocytosis[59] or the specific interaction of each receptor with the known ligands protein S (PROS1) and growth-arrest-specific-gene 6[51,60]. The MERTK AXL[61,62] and TYRO3[61] can modulate cell stiffness of interphase fibroblasts, which leads to elongation of interphase cells. Although this could ultimately play a role in conditioning spindle positioning during mitosis, we observed this phenotype not only upon TYRO3 depletion but also upon depletion of multiple screened genes that did not show a spindle positioning phenotype.

Phosphorylated TYRO3 binds to the SH2 domain of the p85 subunit of PI3K, which catalyzes the addition of phosphate groups to the 3′-OH position on the inositol ring of phosphoinositides (PtdIns), producing PI(3)P, PI(3,4)P2, and PI(3,4,5)P3. Moreover, depletion of PI3K leads to an increase of $PIP_2$ at the cell membrane[12]. We found here that TYRO3 depletion causes an increase of $PIP_2$ at the plasma membrane, together suggesting that the impact of TYRO3 is mediated through an effect on PI3K. It is of interest that NuMA can associate with $PIP_2$ in vitro, which is thought to enhance LGN-independent NuMA association with the cell membrane during anaphase[6,12]. We found here that TYRO3 depletion leads to an increase of NuMA and dynein at the cell cortex in metaphase. Overall, our findings establish that TYRO3 is critical for preventing precocious NuMA deposition at the cell cortex during metaphase spindle positioning, and that this may be mediated, at least in part, through activation of PI3K. Furthermore we found that 34% of cells depleted of the p55 subunit (PIK3R3) exhibited a spindle positioning phenotype in the first round, but this was not deemed to be a hit because the ≥30% cutoff was narrowly missed in the second round of screening (29.4%).

Our findings with TYRO3 may have implications for human disease. TYRO3 is very abundant in the brain compared to other tissues[41] and is an entry receptor for the ZIKA virus[63]. Intriguingly, ZIKA virus infection leads to spindle positioning phenotypes[64], raising the possibility that the impact of the virus on this critical aspect of mitosis may be due to impaired TYRO3

function. TYRO3 is also highly expressed in certain human malignancies, including leiomyosarcoma[65], melanoma[66], leukemia, and lung cancer[67], with TYRO3 genetic knockdown slowing down tumor growth in vitro through unknown mechanisms[65–67]. It will be interesting to consider whether modulation of spindle positioning is involved in this context.

Depletion of GAK, the second strongest hit in our screen, led us to uncover an hitherto unsuspected role for Clathrin in spindle positioning in human cells. The CHC plays a role in spindle positioning in the *C. elegans* zygote via stabilization of the actin–myosin network[68]. By contrast, we show here that depleting GAK or Clathrin leads to a dramatic decrease in astral microtubules emanating from spindle poles. We furthermore establish that this phenotype can be rescued by co-depletion of the microtubule depolymerase MCAK, which is normally inhibited by GTSE1. Since Clathrin loss from central spindle microtubules leads to failed recruitment of GTSE1 to spindle microtubules and renders MCAK hyperactive[49,69], we propose that upon depletion of GAK and, thereby, of Clathrin, astral microtubules are depolymerized due to MCAK being unduly active.

Overall, our work led to the identification of 16 candidate regulators of metaphase spindle positioning, including 11 components that have not been previously associated with such a function. We deciphered the contributions of TYRO3 and GAK to this process, finding that TYRO3 is needed for limiting NuMA distribution, plausibly through $PIP_2$ homeostatic regulation, and that GAK is critical for astral microtubule outgrowth during mitosis, probably through its impact on Clathrin and, thereby, MCAK.

## Methods

**Cell culture.** HeLa mCherry::H2B cells (gift from Arnaud Echard, Institut Pasteur, Paris, France[6]) were used for the screen, whereas for follow-up experiments, we used HeLa Kyoto cells expressing EGFP::α-tubulin and mCherry::H2B (gift from Daniel Gerlich, Institute of Molecular Biotechnology, Vienna, Austria[42]), HeLa cells expressing Lifeact::mCherry and MYH9::GFP (gift from Daniel Müller, Department of Biosystems Science and Engineering, Basel, Switzerland[70]), HeLa Kyoto cells expressing mouse DYNC1H1::GFP (gift from Monica Gotta, University of Geneva, Geneva, Switzerland[71]), regular HeLa cells (ATCC CCL-2), U2OS cells (Sigma, 92022711), and hTERT-RPE-1 cells (ATCC CRL-4000). Cell lines containing different GAK constructs were a gift from David Drubin (University of California, Berkeley, CA, USA[48]). All cells were maintained in high-glucose Dulbecco's modified Eagle's medium (DMEM) with GlutaMAX (Thermo Fisher) medium supplemented with 10% fetal calf serum in a humidified 5% $CO_2$ incubator at 37 °C. For live imaging, the medium was supplemented with 25 mM HEPES (Thermo Fisher) and 1% PenStrep (Thermo Fisher).

**Transfection.** The reverse transfection performed for the screen is described below in the screening pipeline section. For follow-up siRNA experiments, ~100,000 cells were seeded on fibronectin-coated coverslips in 6-well plates and transfected with 20 nM of siRNAs as per the manufacturer's instructions (Invitrogen). In brief, 2.5 μl of 20 μM siRNA in 250 μl OptiMEM and 7.5 μl Lipofectamine RNAi max in 250 μl OptiMEM were incubated in parallel for 5 min, mixed for 20 min, and then added to 2 ml of medium per well. Cells were then incubated for 48 h before fixation and live imaging or collection of cells, depending on the experiment. For plasmid transfections, cells were seeded at 80–90% confluency in 6-well plates. Then, 3 μg of plasmid DNA in 100 μl OptiMEM and 6 μl of Lipofectamine 2000 in 100 ml OptiMEM were incubated in parallel for 5 min, mixed for 20 min, and added to each well. Lipofectamine 3000 was used according to the manufacturer's instructions for transfections of GFP-C1-PLCδ-PH plasmid into HeLa Kyoto cells expressing EGFP::α-tubulin and mCherry::H2B. Transfection efficiency of ~80% was routinely achieved as monitored by GFP staining or by anti-myc staining in the case of the GAK full-length construct. For rescue experiments with GAK constructs, the respective cell lines were transfected with siRNA, except for GAK full-lengths constructs, in which case HeLa cells were siRNA transfected at day 0, and then transfected with a GAK full-length transgene 24 h later, before fixation at 48 h for staining and analysis.

**siRNAs and plasmids.** For the screen, an siRNA library consisting of a mixture of 4 siRNAs corresponding to 1280 kinases, phosphatases, metalloproteases, and G protein-coupled receptors and associated proteins of the human proteome was used (Dharmacon ON-TARGETplus® SMARTpool® siRNA Library, Human Druggable Subsets, G-104675-02 Lot 09159). The negative control for the screen

was scrambled siRNAs (SI03650318, Qiagen), and as the positive control, stealth siRNAs against LGN (sequence: 5′-UAGGAAAUCAUGAUCAAGCAA-3′, Qiagen). For subsequent knockdown of TYRO3 and GAK, two additional siRNAs were used for each gene: siRNAs A (Ambion, silencer select siRNAs, # 802 for TYRO3, 5′- GGUGUGCCAUUUUUCACAGtt-3′; # 118756 for GAK, 5′-CCCGA ACAUUGUCCAGUUUtt-3′) and siRNAs B against the respective 3′-UTRs (Stealth siRNAs, Invitrogen, TYRO3: 5′-CCCACAAUCUGAGCACGCUACCA AA-3′; GAK 5′- CCGUGGUUGUCUGUACAGAAUUAAA-3′). Note that the stealth siRNAs were used for experiments reported in Figs.4–6, and Supplementary Figs. 2 and 4–6. For knockdown of CHC, we used Ambion silencer select siRNAs # s475: 5′-GGUUGCUCUUGGAUtt-3′. For knockdown of MCAK, siRNAs (5′-GAUCCAACGCAGUAAUGGU-3′)[49] were purchased from Ambion. For rescue experiments, the pENTR221 plasmid containing full-length TYRO3 (Life Technologies Ultimate ORF collection provided by EPFL Life Sciences Gene Expression core facility, DCE0026466) was cloned into a pEBTet-EGFP-GW destination vector, thereby creating the EGFP::TYRO3 expression vector using Gateway Technology (Invitrogen). Primers used for EGFP::TYRO3 sequencing were 5′-cgagctgtacaagggtaccgc-3′ and 3′- ccgctagctgccatcaccac-5′. Cells were transfected with GFP-C1-PLCδ-PH plasmid (Addgene, # 21179) for 24 h before live imaging.

**Assay development.** In collaboration with CYTOO (Grenoble), we developed customized 96-well imaging plates with L-shaped micropatterns coated with fibronectin-containing Alexa Fluor 555 separated by a cytophobic region, aiming at minimizing the distance between L-micropatterns to maximize the number of imaged micropatterns per visual field. In the final rendition utilized in the screen, each arm of the L measures 26.5 μm and the distance between micropatterns is 20 μm, as smaller distances led to cell movements between neighboring micropatterns. The final layout yielded 576 micropatterns per visual field. Two visual fields were imaged per well, leading to the visualization of 1152 patterns per condition.

**Screening pipeline.** For reverse transfection, 20 nM (0.1 μl per well) of siRNAs were distributed in 96-well plates (Greiner Cellstar, flat bottom) using a robotic liquid handling system (Biomek F). Negative and positive control siRNAs were distributed in the outer 8 wells on each side of the 96-well plates (see Fig. 1b). Ten microliters of ddH₂O was added to each well for a 5 min incubation of siRNAs, and Lipofectamine RNAi Max 2000 (Invitrogen, 0.3 μl per well) was also incubated in ddH₂O for 5 min (10 μl per well) before 10 μl of the Lipofectamine solution was added to each well for a 20 min incubation. Subsequently, ~6000 HeLa cells expressing mCherry::H2B were seeded per well and incubated for 48 h. CYTOO 96-well imaging plates containing L-shaped micropatterns were pre-incubated at 37 °C for 72 h prior to cell transfer, and another 30 min with DMEM (10% fetal bovine serum (FBS)) added to each well prior to transfer. Cells were washed once in phosphate-buffered saline (PBS) and detached from the well bottom using 40 μl Accutase per well for 8 min (400–600 U/ml, Gibco). To stop the detachment process, DMEM (10% FBS) was then added and the cell density adjusted to 6000/ 100 μl, which was determined to be the optimal number to obtain maximal coverage of micropatterns by single cells. After pipetting up and down using large aperture tips (Finntip, Thermo Fisher) to obtain a suspension of single cells, cells were transferred to the micropatterned imaging 96-well plates. Cells were allowed to settle until full attachment, which took usually 4 h, before starting live imaging, during which plates were sealed (Breathe-Easy sealing membrane, Sigma Z380059).

**Imaging.** The HeLa cell cycle lasts ~24 h and we thus imaged up to that duration to maximize the number of analyzable cells, finding that the percentage of cells dividing is not strikingly different than when imaging for shorter durations (see Supplementary Fig. 1b). Moreover, we found that imaging for longer did not yield more analyzable cells because most cells had divided after that time, resulting in micropatterns being covered by more than one cell. We also determined the optimal frame rate that allowed us to capture at least three images during metaphase and the metaphase–anaphase transition. Since frame rates of 6 and 8 min did not differ significantly with respect to the correct detection of spindle positioning in the negative controls, we chose the latter to minimize data storage (see Supplementary Fig. 1a).

Micropatterned 96-well plates were imaged at 37 °C and 5% $CO_2$ using a ×10 (0.45NA) objective on the GE InCellAnalyzer 2200 microscope equipped with a sCMOS CCD camera and hardware autofocus. Imaging was conducted every 8 min for 24 h, capturing one focal plane with an exposure time of 100 ms, resulting in 180 frames per visual field; 2 visual fields per well were imaged using the Texas red fluorescent channel.

**Data analysis and assay quality.** The imaging data was analyzed initially using TRACMIT[32] and then with the help of a custom KNIME workflow[72]. In the first round, we screened 16 96-well plates and could include all of them in the final analysis based on them having an rSSMD >2 (see below). In the second round, the screening of six plates had to be repeated for the following reasons. In one case, a microscope focus issue was encountered. In another case, the majority of cells were dead. In two other cases, the number of cells to analyze was particularly low. In the two last cases, we repeated the screening for the same reason, but finally did not

include the repeated plates since they harbored even less analyzable cells than in the first pass. Cell fitness was generally worse in the second screening round, as reflected by the slightly lower numbers or analyzable cells, as well as the initial shape of cells, as imaged by bright field microscopy. One plausible explanation is that the second batch of imaging plates contained a different glue that may have yielded a modest level of cytotoxicity.

In order to determine if the data collected from each plate meets quality requirements[35], the rSSMD (strictly standardized median difference) was calculated in each case as follows:

$$\mathrm{rSSMD} = (\mathrm{median}(\mathrm{Cpos}) - \mathrm{median}(\mathrm{Cneg}))/([\mathrm{MAD\ Cpos}])^2 + ([\mathrm{MAD\ Cneg}])^2, \quad (1)$$

where Cpos is the positive control, Cneg is the negative control (Supplementary Fig. 2c), and MAD is the median average deviation. As is apparent, rSSMD uses the median instead of the mean and was developed especially for siRNA-based screens with large variability, for which it is more suitable that the $Z'$- score, which is best suited for more streamlined small-molecule screens[35]. According to the rSSMD-based quality control criteria, values between 1 and 2 are considered as inferior, between 2 and 3 as good and over 3 as excellent (criterion 1b in refs. [2,34]).

In order to decide how many cells should be analyzed per well to get a meaningful result, we performed a sample size calculation based on randomly picked replicates of negative and positive control wells, using plates for which three replicates were available. We used data from cells in 10 wells over three biological replicates and calculated their group means (10 for control siRNA and 25 for LGN siRNA), as well as the mean standard deviation between the screen replicates (5.7 for LGN siRNA and 2.7 for control siRNA). Based on this calculation, at least eight cells per condition should be analyzed to obtain a meaningful result[73]. This could be confirmed by plotting mean and standard deviations of negative and positive control wells as a function of the number of cells per well (Supplementary Fig. 1h).

**HIT selection**. Hits were selected as summarized in Fig. 2 and described in the main text. In the first step, 14 candidates were excluded because no dividing cells were present in either round (*PSMB3, PSMA3, GSK3B, PRSS12, PLK1, PIK3R2, BUB1B*—most likely essential genes or genes critical for cell cycle progression), or at least in one of the two rounds (*FLJ90650, CTSO, DGKH, PRKAG3, FLT3, CTDSP1, PKN3*). The latter group of seven genes originates from different plates, which made it unpractical to repeat the corresponding full plates.

**CRISPR cell lines**. HeLa cells were transfected with CRISPR/Cas9 KO plasmids for TYRO3 and GAK (Santa Cruz sc-401412 and sc-404235, respectively), according to the manufacturer's instructions. After 72 h, transfected cell pools were FACS sorted for GFP-positive cells, which were then singled in 96-well plates. We obtained four viable clones for TYRO3 and three viable clones for GAK.

**Quantitative real-time PCR**. RNAs of cells treated with siRNAs against GAK or TYRO3, as well as from CRISPR/Cas9 cell lines, were extracted by the RNeasy Mini Kit (Qiagen), and complementary DNA (cDNA) was synthesized by the RevertAid First Strand cDNA Synthesis kit (Fermentas). The resulting products were PCR amplified with the following primer pairs (Microsynth): TYRO3 forward primer, 5′-CGCCGCCGCAGGTCT-3′ and TYRO3 reverse primer, 5′-CTGAGGAAGC CGATCCAGTG-3′; GAK forward primer, 5′-GCCGAAGGAGGGTTTGCATT-3′ and GAK reverse primer, 5′-ACTGGACAATGTTCGGGTGG-3′. Reactions were carried out using Power SYBR Green PCR Master Mix (Applied Biosystems, Warrington, UK) on the QuantStudio6 system with 7 flex real-time PCR system software (Applied Biosystems). Melting-curve analyses were performed to verify amplification specificity, and relative quantification performed according to the ΔΔ-CT method[74].

**Western blot analysis**. Whole-cell extracts were separated by sodium dodecyl sulfate-polyacrylamide gel electrophoresis and transferred to a polyvinylidene difluoride membrane according to the manufacturer's protocols (Bio-Rad). After incubation with 5% non-fat milk in TBST (10 mM Tris, pH 8.0, 150 mM NaCl, 0.5% Tween-20) for 60 min, the membrane was washed once with TBST, cut above the 50 kDa band, and both parts incubated with antibodies raised in mouse against GAK (1:1000, ENZO ADI-KAM-ST105-E) or TYRO3 (1:1000, sc 166359) and α-tubulin (1:200 DM1A, Sigma) at 4 °C for 12 h. Membranes were washed three times for 10 min in TBST and incubated with a 1:3000 dilution of horseradish peroxidase-conjugated anti-mouse antibody for 2 h at room temperature in TBST. Blots were washed with TBST three times and developed with the ECL system (Amersham Biosciences), according to the manufacturer's protocol.

**Immunofluorescence**. For immunofluorescence analysis, cells were fixed in −20 °C methanol and washed in PBS 0.05% Triton X-100 (PBST). After blocking in 1% bovine serum albumin in PBST for 1 h at room temperature, cells were incubated with primary antibodies at room temperature for 4 h. After three washes in PBST for 5 min each, cells were incubated with secondary antibodies for 1 h at room temperature, stained with 1 µg/ml Hoechst 33342 (Sigma-Aldrich), washed three times for 5 min in PBST, and mounted. The primary antibodies used were as follows: 1:1000 rabbit anti-POC5[75], 1:1000 rabbit anti-CP110 (127801-AP, ProteinTech Europe), 1:200 mouse anti-α-tubulin (DM1α, Sigma-Aldrich), 1:100

mouse anti-p150^Glued (612709, Transduction Laboratories), 1:100 rabbit anti-NuMA (Santa Cruz, 48773), 1:100 mouse anti-TYRO3 (Santa Cruz, 166359), 1:100 rabbit anti-GAK (Sigma, HPA027463), 1:500 rabbit anti-dynein heavy chain (Santa Cruz, sc-9115), 1:200 mouse anti-actin (Abnova, MAB8172), 1:200 rabbit anti-c-myc (obtained in-house[76]), and 1:500 rabbit anti-CHC (Abcam, 21679). Secondary antibodies were Alexa Fluor 488-coupled anti-mouse and Alexa Fluor 568-coupled anti-rabbit antibodies (Life Technologies), both used at 1:1000 in general, except at 1:500 for NuMA and TYRO3 primary antibodies. The pan-TAM inhibitor BMS777607[44] (Sigma-Aldrich) was added to the medium 6 h before fixation and staining at a concentration of 0.1 µM (in dimethyl sulfoxide).

**Imaging of fixed cells**. Image stacks with a distance of 0.4 µm between optical slices were acquired with a ×63, NA 1.0 oil objective at 526 × 526 pixel minimal resolution on an LSM 700 Zeiss confocal microscope equipped with a charge-coupled device camera (AxioCam MRm), and then processed and analyzed in ImageJ (National Institutes of Health). Relevant optical slices are shown as maximum intensity projections, including for insets of cortical areas. Analysis of lagging chromosomes was done based on the DNA signal using a Zeiss indirect fluorescence microscope and a ×100 objective. Confocal imaging and follow-up image analysis with ImageJ was conducted with the same settings within a series.

**Monitoring spindle positioning on uniform fibronectin**. To monitor spindle positioning, cells were grown on coverslips uniformly coated with fibronectin (Neuvitro, GG-22-fibronectin). After fixation and immunofluorescence, the angle of the metaphase spindle with respect to the fibronectin substratum was determined[64,77]. Cells were stained with Centrin2 or CP110 antibodies to mark spindle poles and counterstained with 1 µg/ml Hoechst 33342 (Sigma-Aldrich) to mark chromosomes. Stacks of images 0.4 µm apart were acquired using confocal microscopy as described below. ImageJ was used to determine distances between spindle poles in $xy$ and $z$, as well as the angle $\alpha$ between a line connecting the two spindle poles and the substratum.

**Spinning disc live microscopy**. HeLa cells expressing EGFP::α-tubulin and mCherry::H2B treated with siRNAs were grown in 35 mm imaging dishes (Ibidi, cat. no. 81156) and imaged at 37 °C using an inverted 574 Olympus IX 81 microscope equipped with a Yokogawa spinning disk CSU-W1 with a ×63 (NA 1.42 U Plan S-Apo) objective and a 16-bit PCO Edge sCMOS camera. Note that the expression levels of EGFP::α-tubulin and mCherry::H2B varied between cells. Images were obtained using 488- and 561-nm solid-state lasers, with an exposure time of 100 ms and a laser power of 11–32%. The imaging medium was as described above. For inhibition of PI3K, 100 µM LY294002[12] (Invitrogen) was added to the medium 2 h before imaging.

**Quantification of cortical signal intensity**. To quantify cortical signal intensities for NuMA and Dynein, two representative 3 × 0.5 µm cortical areas were analyzed per cell side in four different focal planes to ensure broad coverage of the cell cortex. Background-corrected integrated fluorescence intensities were determined for these cortical areas by subtracting the values obtained in a cytoplasmic area of the same size using ImageJ using the following formula:

$$\text{Background corrected integrated FI} = \text{Integrated FI} - (\text{background mean FI} \times \text{area}), \quad (2)$$

where FI is the fluorescence intensity. Finally, one value was derived for each cell by averaging the 16 values obtained in each case.

For actin and GFP-C1-PLCδ, which are distributed more uniformly at the cell cortex, we marked the entire cortex using the ImageJ segmented line tool (width 4 µm). Straightening of this line and running a vertical plot profile helped us obtain the average cortical actin fluorescence intensity and background intensity. Four focal planes were quantified and the results averaged per cell. Actin and GFP-C1-PLCδ fluorescence intensity were expressed as a ratio of cortical/background intensities (Supplementary Figs. 4h, 5h).

**Oscillation index**. HeLa cells expressing EGFP::α-tubulin and mCherry::H2B were imaged with a frame rate of 1 min, and movies were then analyzed using ImageJ. Frames were registered using stackreg to adjust for cell movements between frames. Metaphase angle deviations between frames (frame transitions) were manually determined using the angle tool of ImageJ. Angle deviations of >10°, as well as metaphase plates moving into a different focal plane, were counted as oscillation events. The oscillation index was then determined as the percentage of such events as a function of all frame transitions[12].

**Statistics**. Statistics were performed with GraphPad. We compared two conditions (e.g., ctrl vs. treatment) using unpaired, two-tailed Student's $t$ test, except for samples with unequal variances and sample sizes, in which case we applied a Welch's correction. If the normality test failed for one of the groups within a comparison, then the Mann–Whitney $U$ test was applied instead, as specified in the figure legends in each case.

## Data availability

The entire time-lapse microscopy data set is publicly available in the Image Data Resource (IDR) [https://idr.openmicroscopy.org/] under accession number idr0061. The other data that support the findings of this study are available from the corresponding author upon reasonable request.

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

## Acknowledgements

We are grateful to Sachin Kotak, Petr Strnad, and Fernando R. Balestra for fruitful discussions, to David Drubin, Arnaud Echard, Daniel Gerlich, Monica Gotta, and Daniel Müller for their gift of cell lines. We thank the BioImaging and Optics Platform (BIOP) and the Biomolecular Screening Facility (BSF), both from the School of Life Sciences (EPFL), for imaging and analysis support, as well as to Damiano Banfi (BSF) for writing the code for the genetic algorithm, and Sebastien Degot and Yoran Margaron at CYTOO (Grenoble, France) for help in designing the screening 96-well plates. We also thank Alexandra Bezler, Kerstin Klinkert, and Sachin Kotak for comments on the manuscript. This work was supported by a grant to P.G. from the Swiss National Science Foundation (31003A_155942) and by the European Union FP7 project MEHTRICS (ID 278758), as well as a Mildred-Scheel postdoctoral fellowship to B.W. (110470, German Cancer Aid).

## Author contributions

B.W. and P.G. designed the project; B.W. conducted most experiments with support from C.B.; B.W. and P.G. analyzed the data and wrote the manuscript.

## Additional information

**Competing interests:** The authors declare no competing interests.

**Peer Review Information:** *Nature Communications* thanks Xavier Morin and the other anonymous reviewer(s) for their contribution to the peer review of this work. Peer reviewer reports are available.

