## [Peer Review File · Nature Communications]

Reviewers' comments:

Reviewer #1 (Remarks to the Author):

In this study, Wolf and colleagues develop a live screen for new regulators of mitotic spindle orientation in cultured human cells, using an elegant micro-pattern approach to standardize cell shape and behavior. They screen more than a thousand genes, focusing on kinases, phosphatases, and metalloproteases, resulting in what is arguably the largest screen of its kind for regulators of spindle orientation.

This is an important study: while there is a general feeling in the field that most of the main players of spindle orientation have been identified, the spatial and temporal regulation of their localization remains poorly understood, and many fine regulators remain to be identified. Several previous studies, many of which from this group, have started to show the essential roles of kinases and phosphatases in orchestrating spindle positioning during mitosis progression, but it seems likely that many more are involved. Hence, designing a screen to specifically address the role of the plethora of kinases and phosphatases found in the genome makes a lot of sense, and this study does exactly this. One previous paper (Matsumara et al, 2012, cited in this study) attempted a similar feat, but using a different design, and actually did not find many players. Therefore it makes a lot of sense to try improve on the design and perform a new study of this kind, as proposed here.

This very elegant study, nicely designed and beautifully executed, identifies several new regulators, and tries to elucidate further the mode of action of two of these, the Cyclin G associated Kinase GAK, and Tyrosine-protein kinase receptor TYRO3. The design of this study should be appealing to a broad audience, and the results of the screen, its validation, and the associated resource will be of great interest to the cell biology community.

Besides, the molecular nature of the new players, and the pathways that are identified, open new avenues. In particular, the clathrin-dependent pathway uncovered downstream of GAK is unexpected and will undoubtedly generate follow-up studies.

My main comments and suggestions essentially concern some choices made in the design of the screen and in analyzing the data, which I think require further explanations and discussion from the authors. I also suggest one complementary set of data for the characterization of one of the candidates.

1) The authors choose to use the LGN RNAi orientation defect as their reference point, based on the idea that LGN loss of function has relatively weak phenotypes when measuring spindle orientation defects relative to the substrate or on L-shaped patterns. However, while I think LGN is a good "positive control" in this screen, I am not convinced that it is a good reference point for the cut-off of phenotypes: the reason why LGN has "weak" phenotypes in several contexts (as cited by the authors) is probably because there are several pathways that act in parallel to orient the spindle in these assays, and therefore some degree of redundancy in the system. When considering the LGN complex-dependent orientation pathway, LGN is actually a key player, as it plays a central "mechanical" role in linking microtubules to the cortex. For example, in the "paired-cell assay" used by di Pietro and colleagues, which relies on an LGN complex-dependent orientation pathway, LGN RNAi has a very strong phenotype. One might therefore expect that modulators of the Gai/LGN pathway might have weaker phenotypes than LGN RNAi itself, and therefore that they'd be missed in the present screen if one places the cut-off at the LGN phenotype. Hence the whole analysis is biased by using LGN as a "minimal" reference point. Indeed, as observed by the authors, the SLK RNAi phenotype did not meet their requirement of >30% mis-oriented cells (cells with more than 40° displacement from the 0 position), although SLK was previously described as a regulator of LGN/NuMA dependent orientation (Machicoane et al, JCB 2014) using the same L-shaped patterns and HeLa cells. However, when looking at the data from Machicoane et al, it appears that the phenotypes would also not have been detected with the criteria used in the present study. Actually Machicoane et al used a higher number of cells (>500 cells/condition) and compared the whole distribution, rather than placing a cutoff, and identified SLK as a bona fide regulator with these criteria. Since the whole study here is designed based on the use of LGN as a cut-off, and in particular the number of cells analyzed per condition

was based on this reference point, I suspect that it misses modulators of spindle orientation, and only finds strong players. This methodological choice, and its consequences, should be discussed more thoroughly. In particular, since the authors present the paper as a resource in the discussion, the value and limitations of this resource should be discussed. (Note that this does not reduce the validity of the screen, nor the interest of its results; but it should be discussed).

2) A general remark on angle measurements relative to the substrate and their comparisons. In Supp Figure 2e, angle distributions are shown for the whole cell population, and the comparison is done on the whole distribution (actually I would advise using a Mann Whitney test rather than t-test with Welch correction since it is unlikely that the distribution of angle, which can in principle be anywhere between 0 and 90°, will be normal in these samples). In the rest of the paper, the authors use two bins: angles between 0-10° versus >10°. Why not use a comparison of whole distributions? Using bins reduces the information that is present in the whole distribution, and I am not convinced it is justified for spindle orientation studies.

3) P12/figure4p/q: both GAKdeltaJ and GAKK1 fail to fully rescue the GAK orientation phenotype. However it looks like there is a partial rescue. Is this significant? (when compared to the RNAi alone?). If it is the case, the conclusions of the paragraph p12 on whether the kinase domain is sufficient, and the J domain essential, should be moderated. Again, I think the use of 0-10° versus >10° bins and Student's t-test is not ideal, and that a comparison of the whole distribution of angles between conditions would be more appropriate.

4) P.11: I am surprised that the authors did not monitor PLC δ at the membrane in a greater number of cells to validate or invalidate the increase, as it would be a strong argument in favor of their hypothesis that TYRO3 depletion leads to hypoactivity of PI3K and therefore accumulation of PIP2 and NuMA at the cell cortex.

5) P.15: it is proposed in the discussion that GAK and Clathrin KD cause astral microtubule depolymerization through lack of MCAK inhibition. One might therefore expect that MCAK KD would rescue, to some extent, spindle orientation defects caused by GAK KD. This could in principle be easily tested. A positive result would nicely reinforce the hypothesis (while a negative result would not invalidate it) and complete the study.

Minor points:

- p5. When citing the screen for dynein/dynactin regulators of spindle orientation regulators, they use the wrong reference (di Pietro et al 2016, EMBO Reports instead of di Pietro et al 2017, Curr Biol)

- the cells shown in Fig 4e/f and Supp Fig4h are the same (clathrin staining in Ctrl vs GAK RNAi): probably there are more representative cells than only one per condition, and different cells could be shown! At least it should be mentioned in the legend that the same cells are used in the two figures (and justified?)

- legend to sup figure 2: 5 comparisons are indicated in the legend, only 4 are shown on the figure. Which one does "5 p=0.0116" correspond to?

- Sup Figure 1h: it is not clear from this graph, nor from the text, how one comes to the conclusion that 8 cells is the minimum that should be used.

- Sup figure 2a: what does "% of cells" on the y axis mean? It is not clear from the legend/methods

- Sup figure 2b: I don't understand what "<20 cells" refers to.

Reviewer #2 (Remarks to the Author):

Gonczy and colleagues describe a siRNA-based functional screen to isolate new regulators of mitotic spindle orientations. This elegantly designed screen was conducted in HeLa cells seeded on fibronectin-coated micropatterns. mCherry-H2B cells were visualised by live microscopy and images were analysed by TRACMIT to obtain spindle position (relative to arms of L shape pattern) upon anaphase onset. This work has led to identification of 16 putative candidates; of these 11 are new. The authors validate and evaluate the function of two such proteins: TYRO3 and GAK.

The screen and data analysis have been performed to a high standard, and data presented in the manuscript is of excellent quality. The paper is clearly written and easy to follow, and should appeal to a broad readership.

A previous siRNA screen performed by Matsumara et al assayed the roles of ~700 kinases and related proteins in spindle positioning of HeLa cells (Matsumara, Nat Commun 2012). By contrast, the screen by Gonczy and colleagues interrogates ~1200 kinases, phosphatases and metalloproteases. Although there are clear differences between the two screens (fixed vs live microscopy, coated plastic vs fibronectin micropatterns, unmodified HeLa vs H2B-expressing clonally derived HeLa), I am struck by the lack of overlap between the results. The top hit from the 2012 screen was ABL1, which was validated in HeLa cells and mouse skin; however, ABL1 does not score in the new screen. Likewise, neither GAK nor TYRO3 were among the hits in the 2012 screen (despite being present in the siRNA library). This lack of overlap could be related to the technical differences between the two screens, but it raises the possibility that HeLa cells may not represent a robust experimental system to screen for spindle positioning. It is therefore crucial that the authors perform validation experiments with TYRO3 and GAK in further cell lines using multiple siRNAs. There is an attempt to do this in U2OS cells, but the number of cells scored is disappointingly low (Fig. S2e 9 cells for TYRO3 depletion), and variability between samples appears too great for meaningful statistical comparisons.

Specific points:

1. Despite an obvious effort to provide mechanistic insights into how GAK and TYRO3 act in spindle positioning, these data seem preliminary. The number of cells scored for the various phenotypes appears unusually low especially in Figs 3-4. For instance, the effect on cortical dynein by TYRO3 depletion is small in Fig 3o, and scoring a total of 9 or 10 cells does not seem sufficient for meaningful analysis. Likewise, in Fig 3m, 9 clathrin-depleted cells are scored for astral microtubule phenotype; the effect again is too mild for such a small sample size.

2. In line 239, the authors state that cellular levels of NuMA are unaltered upon TYRO3 depletion. Fig s3i is composed of 4 separate blots to confirm expression levels of NuMA and tubulin in control and TYRO3 knockout cells. I could not find any data on NuMA levels in siRNA-depleted used for IF analysis in Fig. 3. Quantifying total cellular levels of NuMA in mitotic control, TYRO3-depleted and knockout cells will be important to support the effect of TRYO3 on NuMA localisation. Also, quantifying NuMA at spindle poles will provide additional proof that TYRO3 depletion specifically increases cortical NuMA levels.

3. CripsR/Cas9 data on TYRO3 shows only a very mild defect, which is surprising. The authors suggest that this could be due to compensation; to prove that it is indeed compensation, and not off-target effects, the knockout cells could be treated by siRNA. Rescue experiments are viable alternatives. Also, NuMA localisation should be shown in TYRO3 knockout cells.

4. As for GAK, the authors convincingly show that it acts with clathrin in spindle positioning, potentially through astral microtubules. In their discussion they mention recent results from the Bird lab on how GTSE1 (which is dependent on clathrin for its spindle localisation) suppresses MCAK activity, a process important for astral microtubule stability. To confirm if the cause of abnormal spindle positioning in GAK-deficient cells is indeed astral microtubule loss, MCAK could be depleted in GAK knockout cells (alternatively, low dose taxol treatment may be used), and spindle position could be analysed. Given the requirement for the clathrin-binding domain of GAK in spindle positioning, it is highly likely that the downstream pathway will involve astral microtubule stability and MCAK. Including these experiments would make the manuscript feel more complete and potentially improve its impact.

5. In lines 198-199, it is stated that TYRO3 and GAK depletion can occasionally lead to lagging chromosomes, and that such cells are excluded from analysis. In a report by the Medema group,

GAK depletion generated a profound chromosome misalignment phenotype (40-50% of cells). Could the authors clarify what 'occasionally' means? Is not there a risk that the cells excluded from the analysis due to lagging chromosomes are the best depleted ones? Furthermore, because siRNA-treated cells may vary in their GAK expression, and off target effects cannot be excluded, it would be useful to perform image analysis of astral microtubule and clathrin levels in GAK knockouts.

Minor points

1. Spindle oscillations should be clearly defined in main text. Currently it is stated that excessive metaphase spindle movements occur in TYRO3 depleted cells, but it is not defined as oscillation (line 233).
2. New siRNA as a term is confusing in figures. Instead, siRNA A/B can be introduced in main text.
3. Line 252: correct IP3K to PI3K
4. Fig 2d and e: cell numbers are missing.
5. I cannot find a fifth p value in Fig S2e. What does $5 p = 0.0116$ in figure legend refer to?

Reviewer #3 (Remarks to the Author):

Wolf et al. describe a live-imaging based siRNA screen for regulators of spindle positioning in HeLa cells. From amongst 1280 kinases, phosphatases, metalloproteases and related proteins, they identify TYRO3, GAK and several other genes (total 16) that alter spindle positioning. TYRO3 and GAK were validated by additional CRISPR mutagenesis. The paper is clearly written, the screen is innovative and the data are strong. The use of LGN as control is important and useful. Spindle positioning and its effects on mammalian cell division is an important, if controversial, fundamental question.

That said, the primary concerns stem from the described phenotype of TYRO3 knockout mouse. I am concerned whether the candidates identified or results described have any biological relevance or are merely artifacts of the culture system and cells used.

- 1) The lack of any observable effects in TYRO3 knockout mice raises significant concern about the biological relevance of the HeLa cell on L-shaped substrate model used for spindle positioning. If spindle positioning is important for development and homeostasis, and TYRO3 can be ascribed an important role, the lack of any significant developmental or homeostatic phenotype in knockout mice is disconcerting. Even Tyro3, Axl, Mertk triple knockout mice are viable. The lack of any biological effect raises significant concern regarding the model used for the study.
- 2) TYRO3 is found in vertebrates and lacks invertebrate homologs. Spindle positioning regulators are expected to be evolutionarily conserved.
- 3) Other concerns relate to lack of a pathway that can be identified in the screen. From amongst 1280 kinases and phosphatases, one would expect a few that can be integrated in a functional pathway. Other than MLCK and STK17B, this doesn't appear obvious. If indeed PI3-K signaling downstream TYRO3 is implicated in the phenotype, wouldn't it be logical to expect PI3-K subunits or Pten to show up in the screen?
- 4) Similarly, it is surprising that the highly homologous AXL or MERTK (if expressed in HeLa cells), which also signals via PI3-K, did not show up in the screen. The expression should be checked.
- 5) Identifying domains or motifs in TYRO3 and a more mechanistic understanding about how TYRO3 regulates spindle positioning seems absolutely crucial to support the findings described. Rescue experiments (with WT and mutant TYRO3) are essential to provide confidence in the

conclusions. Is kinase function of TYRO3 required? Would small molecule TYRO3 kinase inhibitors affect spindle orientation?

6) Are TYRO3 ligands important for the observed effect?

7) Protein expression (and if possible localization) should be validated more carefully. For protein expression, immunoblotting should be used (to check for bands of predicted molecular weight). As indicated by the authors, centrosomal (and mid-body) localization is often an artefact.

8) STK17B, as well as TYRO3, have been linked with apoptosis and sensing of apoptotic cells, respectively. Can interaction of cells with apoptotic bodies alter their 'polarity' even when interacting with the L-shaped scaffold?

9) P2RY5 is neither a kinase, phosphatase or metalloprotease. Neither is TAAR2 or CXCR6. G-protein coupled receptors should be included as a category.

10) A bit strange that kinases such as LLGL and aPKC, with previously described roles in spindle asymmetric division, are not detected in the assay.

Point-by-point response to the reviewers' comments

We are glad that the reviewers found our work to be “nicely designed and beautifully executed” (reviewer 1), of “excellent quality” (reviewer 2), and “the screen innovative and the data strong” (reviewer 3). In addition, we thank the reviewers for having raised a number of important points, which we addressed in full in revising the manuscript, as detailed point by point below.

Reviewer 1

In this study, Wolf and colleagues develop a live screen for new regulators of mitotic spindle orientation in cultured human cells, using an elegant micro-pattern approach to standardize cell shape and behavior. They screen more than a thousand genes, focusing on kinases, phosphatases, and metalloproteases, resulting in what is arguably the largest screen of its kind for regulators of spindle orientation.

This is an important study: while there is a general feeling in the field that most of the main players of spindle orientation have been identified, the spatial and temporal regulation of their localization remains poorly understood, and many fine regulators remain to be identified. Several previous studies, many of which from this group, have started to show the essential roles of kinases and phosphatases in orchestrating spindle positioning during mitosis progression, but it seems likely that many more are involved. Hence, designing a screen to specifically address the role of the plethora of kinases and phosphatases found in the genome makes a lot of sense, and this study does exactly this. One previous paper (Matsumara et al, 2012, cited in this study) attempted a similar feat, but using a different design, and actually did not find many players. Therefore it makes a lot of sense to try improve on the design and perform a new study of this kind, as proposed here.

This very elegant study, nicely designed and beautifully executed, identifies several new regulators, and tries to elucidate further the mode of action of two of these, the Cyclin G associated Kinase GAK, and Tyrosine-protein kinase receptor TYRO3. The design of this study should be appealing to a broad audience, and the results of the screen, its validation, and the associated resource will be of great interest to the cell biology community.

Besides, the molecular nature of the new players, and the pathways that are identified, open new avenues. In particular, the clathrin-dependent pathway uncovered downstream of GAK is unexpected and will undoubtedly generate follow-up studies.

My main comments and suggestions essentially concern some choices made in the design of the screen and in analyzing the data, which I think require further explanations and discussion from the authors. I also suggest one complementary set of data for the characterization of one of the candidates.

1) The authors choose to use the LGN RNAi orientation defect as their reference point, based on the idea that LGN loss of function has relatively weak phenotypes when measuring spindle orientation defects relative to the substrate or on L-shaped patterns. However, While I think LGN is a good “positive control” in this screen, I am not convinced that it is a good reference point for the cut-off of phenotypes: the reason why LGN has “weak” phenotypes in several contexts (as cited by the authors) is probably because there are several pathways that act in parallel to orient the spindle in these assays, and therefore some degree of redundancy in the system. When considering the LGN complex-dependent orientation pathway, LGN is actually a key player, as it plays a central “mechanical” role in linking microtubules to the cortex. For example, in the “paired-cell assay” used by di Pietro and

colleagues, which relies on an LGN complex-dependent orientation pathway, LGN RNAi has a very strong phenotype. One might therefore expect that modulators of the Gai/LGN pathway might have weaker phenotypes than LGN RNAi itself, and therefore that they'd be missed in the present screen if one places the cut-off at the LGN phenotype. Hence the whole analysis is biased by using LGN as a "minimal" reference point. Indeed, as observed by the authors, the SLK RNAi phenotype did not meet their requirement of >30% mis-oriented cells (cells with more than 40° displacement from the 0 position), although SLK was previously described as a regulator of LGN/NuMA dependent orientation (Machicoane et al, JCB 2014) using the same L-shaped patterns and HeLa cells. However, when looking at the data from Machicoane et al, it appears that the phenotypes would also not have been detected with the criteria used in the present study. Actually Machicoane et al used a higher number of cells (>500 cells/condition) and compared the whole distribution, rather than placing a cutoff, and identified SLK as a bona fide regulator with these criteria. Since the whole study here is designed based on the use of LGN as a cut-off, and in particular the number of cells analyzed per condition was based on this reference point, I suspect that it misses modulators of spindle orientation, and only finds strong players. This methodological choice, and its consequences, should be discussed more thoroughly. In particular, since the authors present the paper as a resource in the discussion, the value and limitations of this resource should be discussed. (Note that this does not reduce the validity of the screen, nor the interest of its results; but it should be discussed).

> We fully concur with the reviewer: *bona fide* regulators of spindle positioning that exhibit phenotypes weaker than those observed following LGN depletion would not have been retained in our final list of candidates. However, utilizing stringent selection criteria was a deliberate choice on our part, which enabled us to focus on those components with the strongest contribution to spindle positioning, as exemplified by TYRO3 and GAK. Moreover, doing so enabled us to decrease the occurrence of false positives. Importantly, in addition, components that contribute in a weaker manner to spindle positioning can nevertheless be identified in our data set by mining Table S1, which reports the results from the entire screen. We mention these points in a fuller manner in the Discussion section of the revised manuscript (p. 14).

As for the comparison with the Machicoane et al. publication: since we do not have access to the corresponding raw data, we cannot state with certainty whether applying the cutoff chosen in our study would have permitted the identification of SLK as a spindle positioning regulator in the earlier data set. As an alternative means of comparison, we investigated the impact of displaying the whole distribution of angles according to Machicoane et al. for two wells from two rounds of screening for the negative control, the positive control and the SLK siRNA condition from our data set (Rebuttal Fig. 1a). Importantly, we found that whereas the negative and positive control distribution differed significantly ($p = 0.002$, Mann Whitney U test, $n > 100$ cells in each group, 2 biological replicates), there was no significant difference between the negative control and the SLK depletion distribution (Mann Whitney U test, $p = 0.92$). Therefore, the same conclusion would be reached for SLK in our data set by using the cutoff and the output of the whole angle distribution. Moreover, we compared our SLK siRNA results displayed in this manner with the relevant published figure panel from Machicoane et al. (shown in Rebuttal Fig. 1b). This comparison shows that whereas Machicoane et al. found a significantly altered angle distribution upon SLK siRNA, which was not the case in our work. Several reasons could explain this apparent discrepancy. First, our analysis was carried out within a large scale screen conducted in 96 well plates, whereas in the former

study one set of siRNAs was applied at a time on coverslips, enabling also the scoring of a higher number of cells. Second, different siRNAs were utilized in the two studies.

2) A general remark on angle measurements relative to the substrate and their comparisons. In Supp Figure 2e, angle distributions are shown for the whole cell population, and the comparison is done on the whole distribution (actually I would advise using a Mann Whitney test rather than t-test with Welch correction since it is unlikely that the distribution of angle, which can in principle be anywhere between 0 and 90°, will be normal in these samples). In the rest of the paper, the authors use two bins: angles between 0-10° versus >10°. Why not use a comparison of whole distributions? Using bins reduces the information that is present in the whole distribution, and I am not convinced it is justified for spindle orientation studies.

> We thank the reviewer for this comment, which relates to the first point made above. The full distribution was always utilized for statistical analysis using GraphPad prism; if samples did not pass testing for normal distribution, the Mann Whitney U test was used. We also carefully recalculated normality for all used angle distributions and adapted the used statistical test (Mann Whitney U vs. t-test with Welch correction) when needed. We state this more explicitly in all the Figure legends and in the Methods section of the revised manuscript (p. 26). Nevertheless, we chose to retain the current binary mode for spindle angles (i.e. 0-10 or >10°) for all analysis done on uniform fibronectin, to facilitate handling and displaying of our large data set. Furthermore, we note that spindle angles on uniform fibronectin coated coverslips are of 45° at most, such that the distribution is not as spread as on the L-shaped micropatterns.

3) P12/figure4p/q: both GAKdeltaJ and GAKK1 fail to fully rescue the GAK orientation phenotype. However it looks like there is a partial rescue. Is this significant? (when compared to the RNAi alone?). If it is the case, the conclusions of the paragraph p12 on whether the kinase domain is sufficient, and the J domain essential, should be moderated. Again, I think the use of 0-10° versus >10° bins and Student's t-test is not ideal, and that a comparison of the whole distribution of angles between conditions would be more appropriate.

> We are grateful to the reviewer for bringing up this important point. Indeed, as anticipated by her/him, statistical analysis established that both ΔJ GAK and GAK kinase domain constructs exhibited partial rescue of the GAK siRNA phenotype ($p < 0.01$ in both cases using Mann Whitney U test of the full distribution). Although dimerization of these constructs with residual endogenous GAK might explain in part such partial rescue, these findings suggest that the kinase domain also contributes to robust GAK-mediated spindle positioning. The text of the revised manuscript as well as Fig. 4 p,q has been altered accordingly (p. 13).

4) P.11: I am surprised that the authors did not monitor PLC δ at the membrane in a greater number of cells to validate or invalidate the increase, as it would be a strong argument in favor of their hypothesis that TYRO3 depletion leads to hypoactivity of PI3K and therefore accumulation of PIP₂ and NuMA at the cell cortex.

> The reason for not including more cells in the initial analysis reflected the difficulty in finding conditions in which cells transfected with TYRO3 siRNA plus the plasmid expressing GFP-C1-PLC δ would survive when filmed during cell division. Prompted by the comment of the reviewer, we optimized the transfection conditions and used HeLa Kyoto cells expressing EGFP:: α -tubulin and mCherry::H2B, which enabled us to substantially increase the data set. Thus, we filmed and analyzed 24 control and 18 TYRO3 depleted cells in a new set of experiments. Importantly, this analysis demonstrates that PIP₂ levels at the cell membrane

are indeed significantly increased upon TYRO3 depletion. This new data is reported in Supplemental Figure 4g-h and mentioned in the revised text (p. 11).

5) P.15: it is proposed in the discussion that GAK and Clathrin KD cause astral microtubule depolymerization through lack of MCAK inhibition. One might therefore expect that MCAK KD would rescue, to some extent, spindle orientation defects caused by GAK KD. This could in principle be easily tested. A positive result would nicely reinforce the hypothesis (while a negative result would not invalidate it) and complete the study.

> We thank this reviewer, as well as reviewer 2, for suggesting this interesting experiment, which we performed as recommended. Importantly, we found that treating cells depleted of GAK with siRNAs targeting MCAK indeed rescued the spindle positioning phenotype. Furthermore, we found that depletion of MCAK restores astral microtubules to cells depleted of GAK (example in Fig. S4l). Together, these new results validate the hypothesis according to which GAK ensures a robust astral microtubule network through negative regulation of MCAK and is reported in the new Supplemental Figure 4k, as well as on p. 13 of the revised manuscript. See also response to point 4 of reviewer 2.

Minor points:

- p5. When citing the screen for dynein/dynactin regulators of spindle orientation regulators, they use the wrong reference (di Pietro et al 2016, EMBO Reports instead of di Pietro et al 2017, Curr Biol)

> We thank the reviewer for spotting this error. The reference has been corrected.

- the cells shown in Fig 4e/f and Supp Fig4h are the same (clathrin staining in Ctrl vs GAK RNAi): probably there are more representative cells than only one per condition, and different cells could be shown! At least it should be mentioned in the legend that the same cells are used in the two figures (and justified?)

> We thank the reviewer for this comment. Consequently, we changed Supplemental Figure 4i to show different cells than those shown in Figure 4e/f.

- legend to sup figure 2: 5 comparisons are indicated in the legend, only 4 are shown on the figure. Which one does "5 p=0.0116" correspond to?

We thank the reviewer for having spotted this typo, which we corrected.

Sup Figure 1h: it is not clear from this graph, nor from the text, how one comes to the conclusion that 8 cells is the minimum that should be used.

> We agree with the reviewer and added statistical analysis of the differences between positive and negative controls (p-values are now displayed on the right Y-axis, Fig. S1h).

- Sup figure 2a: what does "% of cells" on the y axis mean? It is not clear from the legend/methods

> We apologize for not having been sufficiently clear on this point in the initial submission. We have altered the designations on the X and Y axes, and updated the figure legend to hopefully better explain what the graph reports.

- Sup figure 2b: I don't understand what "<20 cells" refers to.

> We thank the reviewer for having detected this typo, which was a carry over from an earlier version of this figure panel. This has been corrected.

Reviewer #2 (Remarks to the Author):

Gonczy and colleagues describe a siRNA-based functional screen to isolate new regulators of mitotic spindle orientations. This elegantly designed screen was conducted in HeLa cells seeded on fibronectin-coated micropatterns. mCherry-H2B cells were visualised by live microscopy and images were analysed by TRACMIT to obtain spindle position (relative to arms of L shape pattern) upon anaphase onset . This work has led to identification of 16 putative candidates; of these 11 are new. The authors validate and evaluate the function of two such proteins: TYRO3 and GAK.

The screen and data analysis have been performed to a high standard, and data presented in the manuscript is of excellent quality. The paper is clearly written and easy to follow, and should appeal to a broad readership.

A previous siRNA screen performed by Matsumara et al assayed the roles of ~700 kinases and related proteins in spindle positioning of HeLa cells (Matsumara, Nat Commun 2012). By contrast, the screen by Gonczy and colleagues interrogates ~1200 kinases, phosphatases and metalloproteases. Although there are clear differences between the two screens (fixed vs live microscopy, coated plastic vs fibronectin micropatterns, unmodified HeLa vs H2B-expressing clonally derived HeLa), I am struck by the lack of overlap between the results. The top hit from the 2012 screen was ABL1, which was validated in HeLa cells and mouse skin; however, ABL1 does not score in the new screen. Likewise, neither GAK nor TYRO3 were among the hits in the 2012 screen (despite being present in the siRNA library). This lack of overlap could be related to the technical differences between the two screens, but it raises the possibility that HeLa cells may not represent a robust experimental system to screen for spindle positioning. It is therefore crucial that the authors perform validation experiments with TYRO3 and GAK in further cell lines using multiple siRNAs. There is an attempt to do this in U2OS cells, but the number of cells scored is disappointingly low (Fig. S2e 9 cells for TYRO3 depletion), and variability between samples appears too great for meaningful statistical comparisons.

>In her/his opening remarks, this reviewer wondered about differences between the Matsumara et al. work and the present study. She/he also wondered whether HeLa cells are a robust experimental system to screen for spindle positioning.

>We thank the reviewer for bringing up these points. We begin by considering differences in experimental design and screen execution between the Matsumara et al. work and the present study. First, Matsumara et al. used the silencer Kinase siRNA library from Ambion targeting 719 genes, with 3 distinct siRNAs/gene that were analyzed separately, whereas we used the Dharmacon ON-TARGETplus® SMARTpool® siRNA Library against 1280 kinases, phosphatases, metalloproteinases and associated proteins with a mix of 4 siRNAs per gene. Second, a double thymidine block was utilized by Matsumara et al. to synchronize cells, whereas we worked with unsynchronized cells. Third, Matsumara et al. plated cells on 96 well plates coated with fibronectin and scored spindle orientation with respect to that substratum in fixed specimens, whereas we used live imaging to monitor the process with respect to L-shaped micropatterns. Fourth, Matsumara et al. performed one screening round for the majority of genes, conducting repeats only with potential hits from the first round. As a

case in point, TYRO3 was screened in only one round and did not pass their empirical cutoff, despite significant p-values with all three siRNAs. GAK was screened in only one round as well, exhibiting a significant p-value for only 1/3 siRNAs. Potentially as a result of experimental design and screen execution, other known regulators of spindle positioning, including PLK2, PLK4, ILK, MARK1, STK11, SLK were also not identified by Matsumara et al. (see Table S2). Taken together, the above considerations likely explain why the overlap between that work and our study is only partial. We mention some of these points in the Discussion section of the revised manuscript (p. 14).

As for the second point brought up by this reviewer in her/his opening remarks: we have conducted additional experiments to increase the number of U2OS cells depleted of TYRO3 and GAK that were analyzed, as well as to investigate the consequences of their depletion in non-transformed hTERT-RPE-1 cells. These new results are shown in Supplemental Figure 2e - g and firmly establish that TYRO3 and GAK are required for proper spindle positioning in several transformed and non-transformed human cell lines.

Specific points:

1. Despite an obvious effort to provide mechanistic insights into how GAK and TYRO3 act in spindle positioning, these data seem preliminary. The number of cells scored for the various phenotypes appears unusually low especially in Figs 3-4. For instance, the effect on cortical dynein by TYRO3 depletion is small in Fig 3o, and scoring a total of 9 or 10 cells does not seem sufficient for meaningful analysis. Likewise, in Fig 3m, 9 clathrin-depleted cells are scored for astral microtubule phenotype; the effect again is too mild for such a small sample size.

> Prompted by the comment of the reviewer, we increased the sample size for several experiments. In particular, the numbers of TYRO3 depleted cells analyzed for cortical dynein distribution has increased from (n metaphase/anaphase 8/11 to 18/18 (ctrl) as well as from 9/11 to 23/22 (TYRO3 siRNA) in the revised manuscript (Figure 3o). Likewise, instead of 7 (ctrl), 9 (clathrin siRNA), 8 (GAK siRNA) cells initially analyzed, 52 (ctrl), 29 (clathrin siRNA), 21 (GAK siRNA) as well as 37 (GAK CRISPR # 1) cells are now reported in the scoring of the astral microtubule phenotype (Fig. 4m). Important, these additional experiments fully confirm the effects already observed with lower sample sizes.

2. In line 239, the authors state that cellular levels of NuMA are unaltered upon TYRO3 depletion. Fig s3i is composed of 4 separate blots to confirm expression levels of NuMA and tubulin in control and TYRO3 knockout cells. I could not find any data on NuMA levels in siRNA-depleted used for IF analysis in Fig. 3. Quantifying total cellular levels of NuMA in mitotic control, TYRO3-depleted and knockout cells will be important to support the effect of TRYO3 on NuMA localisation. Also, quantifying NuMA at spindle poles will provide additional proof that TYRO3 depletion specifically increases cortical NuMA levels.

> We agree with the reviewer. In fact, the quantification of spindle pole NuMA was already shown in Supplemental Figure 4g in control and TYRO3siRNA conditions (now Supplemental Figure 4c). We furthermore now quantified total cellular NuMA in mitotic cells using immunofluorescence analysis. The corresponding results are shown in Supplemental Figure 4b and establish that there is no significant different in overall NuMA levels upon TYRO3 depletion.

3. CripsR/Cas9 data on TYRO3 shows only a very mild defect, which is surprising. The authors suggest that this could be due to compensation; to prove that it is indeed

compensation, and not off-target effects, the knockout cells could be treated by siRNA. Rescue experiments are viable alternatives. Also, NuMA localisation should be shown in TYRO3 knockout cells.

> We performed two sets of experiments prompted by this comment. First, we tested whether spindle positioning defects in the TYRO3 CRISPR/Cas9 cell line could be rescued by overexpression of EGFP::TYRO3, and found this to be the case indeed (new Supplemental Figure 3n). Furthermore, we tested whether adding TYRO3 siRNAs would worsen the spindle positioning phenotype of TYRO3 CRISPR/Cas9 cells, which we found not to be the case (new Supplemental Figure 3n). Together, these findings firmly establish that the milder phenotype of TYRO3 CRISPR/Cas9 cells is not due to off-target effects, as mentioned on p. 9 of the revised manuscript.

Furthermore, as requested by the reviewer, we now show NuMA distribution in TYRO3 CRISPR/Cas9 cells (new Fig. S3q).

4. As for GAK, the authors convincingly show that it acts with clathrin in spindle positioning, potentially through astral microtubules. In their discussion they mention recent results from the Bird lab on how GTSE1 (which is dependent on clathrin for its spindle localisation) suppresses MCAK activity, a process important for astral microtubule stability. To confirm if the cause of abnormal spindle positioning in GAK-deficient cells is indeed astral microtubule loss, MCAK could be depleted in GAK knockout cells (alternatively, low dose taxol treatment may be used), and spindle position could be analysed. Given the requirement for the clathrin-binding domain of GAK in spindle positioning, it is highly likely that the downstream pathway will involve astral microtubule stability and MCAK. Including these experiments would make the manuscript feel more complete and potentially improve its impact.

> We thank this reviewer, as well as reviewer 1, for suggesting this interesting experiment, which we performed as recommended. Importantly, we found that treating cells depleted of GAK with siRNAs targeting MCAK indeed rescued the spindle positioning phenotype. This result validates the hypothesis according to which GAK ensures a robust astral microtubule network through negative regulation of MCAK and is reported in the new Supplemental Figure 4j, l, as well as on p. 13 of the revised manuscript. See also response to point 5 of reviewer 1.

5. In lines 198-199, it is stated that TYRO3 and GAK depletion can occasionally lead to lagging chromosomes, and that such cells are excluded from analysis. In a report by the Medema group, GAK depletion generated a profound chromosome misalignment phenotype (40-50% of cells). Could the authors clarify what 'occasionally' means? Is not there a risk that the cells excluded from the analysis due to lagging chromosomes are the best depleted ones? Furthermore, because siRNA-treated cells may vary in their GAK expression, and off target effects cannot be excluded, it would be useful to perform image analysis of astral microtubule and clathrin levels in GAK knockouts.

> We apologize for not having been sufficiently clear on this point. The reviewer is of course correct that excluding cells with lagging chromosomes from our analysis will bias the results away from the most severe depletion phenotypes. However, we prefer to follow this prudent approach to avoid potentially confounding issues stemming from chromosome segregation defects. We explain this point in a more explicit manner in the revised manuscript (p. 9). Moreover, we have now quantified the percentage of cells exhibiting misaligned chromosomes following TYRO3 and GAK depletion by siRNAs, as well as in the corresponding CRISPR/Cas9 cell lines. The results are shown in Fig. S2i, and mentioned in the revised manuscript (p. 12).

Furthermore, as suggested by the reviewer, we performed quantification of astral microtubule distribution and of Clathrin distribution in GAK CRISPR/Cas9 cells (new Figure 4m and new Supplemental Figure 4i), which fully confirmed the findings made initially with siRNA-mediated depletion experiments.

Minor points

1. Spindle oscillations should be clearly defined in main text. Currently it is stated that excessive metaphase spindle movements occur in TYRO3 depleted cells, but it is not defined as oscillation (line 233).

> We followed the reviewer's suggestion and changed the term to "oscillations" in line 233 and throughout the manuscript.

2. New siRNA as a term is confusing in figures. Instead, siRNA A/B can be introduced in main text.

> We also followed this proposal and now term the other siRNAs used for confirmation siRNAs A and B.

3. Line 252: correct IP3K to PI3K

> We thank the reviewer and corrected this typo.

4. Fig 2d and e: cell numbers are missing.

We thank the reviewer and added cell numbers in the legend of Figure 2.

5. I cannot find a fifth p value in Fig S2e. What does 5 p = 0.0116 in figure legend refer to?

> We thank the reviewer for having detected this mistake. Since we changed this figure completely to integrate the U2OS and hTERT-RPE-1 data, this issue is no longer pertinent.

Reviewer #3 (Remarks to the Author):

Wolf et al. describe a live-imaging based siRNA screen for regulators of spindle positioning in HeLa cells. From amongst 1280 kinases, phosphatases, metalloproteases and related proteins, they identify TYRO3, GAK and several other genes (total 16) that alter spindle positioning. TYRO3 and GAK were validated by additional CRISPR mutagenesis. The paper is clearly written, the screen is innovative and the data are strong. The use of LGN as control is important and useful. Spindle positioning and its effects on mammalian cell division is an important, if controversial, fundamental question.

That said, the primary concerns stem from the described phenotype of TYRO3 knockout mouse. I am concerned whether the candidates identified or results described have any biological relevance or are merely artifacts of the culture system and cells used.

1) The lack of any observable effects in TYRO3 knockout mice raises significant concern about the biological relevance of the HeLa cell on L-shaped substrate model used for spindle positioning. If spindle positioning is important for development and homeostasis, and TYRO3 can be ascribed an important role, the lack of any significant developmental or homeostatic phenotype in knockout mice is disconcerting. Even Tyro3, Axl, Mertk triple knockout mice are

viable. The lack of any biological effect raises significant concern regarding the model used for the study.

> The reviewer is concerned about the fact that TYRO3, AXL, MERTK triple knock out mice are viable (Li et al. 2013). As discussed by others (Lemke 2013), although TAM receptors are dispensable during embryogenesis, they are crucial for adult tissue homeostasis, for instance in decreasing inflammatory processes (Rothlin et al. 2015). Whether centrosome and/or spindle positioning may be altered in such adult contexts is an interesting avenue for future investigations. Moreover, redundancy among spindle positioning pathways may also explain why effects observed in tissue culture cells are not mirrored fully *in vivo*. Such a phenomenon has been observed in many other instances, including for Cyclin-dependent-kinases, several of which are essential for cell cycle progression in tissue culture cells but not in mice owing to robust compensatory mechanisms operating *in vivo* (Rane et al. 1999; Berthet et al. 2003; Ortega et al. 2003). The above points help understand why our findings with TYRO3 are of importance even if mice can live without this protein and are discussed explicitly in the revised manuscript (p. 15).

Furthermore, we have tested the impact of TYRO3 and GAK depletion in different cell lines, including now untransformed hTERT-RPE-1 cells, as described also in our response to Reviewer 2. These results are shown in the new Supplemental Figure 2e-g and, together with the increased numbers for U2OS cells, establish that the observed phenotypes are not limited to HeLa cells.

2) TYRO3 is found in vertebrates and lacks invertebrate homologs. Spindle positioning regulators are expected to be evolutionarily conserved.

> We respectfully disagree with the reviewer on this point. One of the very purposes of the screen was to discover novel spindle positioning regulators, including components not conserved across all metazoan species, and which may therefore have been missed in screens performed in flies or nematodes for instance. Many cellular processes are endowed with additional layers of regulation in vertebrates compared to invertebrates. We note also that a TAM-like receptor is present in prevertebrate urochordates (Kulman et al. 2006; Lemke and Rothlin 2008), raising the possibility that the role of TYRO3 uncovered here is not limited to vertebrates. We altered the wording to explain more explicitly that the present screen was expected to identify some vertebrate-specific components, as exemplified by TYRO3 and GAK (p. 15).

3) Other concerns relate to lack of a pathway that can be identified in the screen. From amongst 1280 kinases and phosphatases, one would expect a few that can be integrated in a functional pathway. Other than MLCK and STK17B, this doesn't appear obvious. If indeed PI3-K signaling downstream TYRO3 is implicated in the phenotype, wouldn't it be logical to expect PI3-K subunits or Pten to show up in the screen?

> We thank the reviewer for bringing up these points. We trust that using the present resource for pathway detection will be a powerful means for discovering networks contributing to spindle positioning. For instance, connections are already apparent between the 16 candidate hits, as can be seen in the STRING analysis shown in Rebuttal Figure 1c. Furthermore, mining the data in Table S1 in its entirety is expected to lead to the detection of further interactions among spindle positioning candidates.

As for PI3-K subunits, we found that 34% of cells depleted of the p55 subunit (PIK3R3) exhibited a spindle positioning phenotype in the first round, but this was not called a hit because the 30% cutoff was not met in the second round of screening (29.4%). We found

similar results for PIK3R4 (33.3% and 26.3%), while for PIK3R2 (p85) no dividing cells could be analyzed after siRNA treatment. PIK3R1 was not a hit with 10% and 20.6% spindle positioning phenotypes in two rounds of screening. Given known redundancies among PI3-K subunits, one would not necessarily expect all single subunits to be essential for proper spindle positioning. As for PTEN, perhaps the lack of an apparent role in spindle positioning in the present screen (22.2%, phenotype in first round and 11.6% in second round) reflects the fact that it regulates PI3K without the implication of G-protein coupled receptors. We discuss those findings in the new version of the manuscript (p. 15).

4) Similarly, it is surprising that the highly homologous AXL or MERTK (if expressed in HeLa cells), which also signals via PI3-K, did not show up in the screen. The expression should be checked.

> Depletion of AXL indeed did not yield a spindle positioning phenotype in either screening round (23.6% phenotype in first round and 16% in second round). By contrast, between the two rounds of screening, 28% (22% and 34%) of cells depleted of MERTK exhibited a spindle positioning phenotype, thus slightly missing our stringent cutoff.

All three TAM are indeed expressed in HeLa cells (Linger et al. 2008). Although many of their functions are overlapping (Linger et al. 2008), single TAMs also have unique functions in some cases, as exemplified by the role of MERTK in phagocytosis (D'Cruz et al. 2000) or the very specific interaction of each receptor with GAS6 and PROS1 (Tsou et al. 2014). The above points are mentioned in the Discussion of the revised manuscript (p. 15).

5) Identifying domains or motifs in TYRO3 and a more mechanistic understanding about how TYRO3 regulates spindle positioning seems absolutely crucial to support the findings described. Rescue experiments (with WT and mutant TYRO3) are essential to provide confidence in the conclusions. Is kinase function of TYRO3 required? Would small molecule TYRO3 kinase inhibitors affect spindle orientation?

> As already stated in response to point 3 of Reviewer 2, we have added rescue data for the TYRO3 knock out cell lines (new Supplemental Figure 3n). Furthermore, we used the small molecule inhibitor BMS 777607 to test whether TYRO3 kinase activity is required for proper spindle positioning. BMS 777607 is the most specific commercially available TYRO3 inhibitor (IC₅₀ = 4.3 nM), although it also blocks AXL (IC₅₀ = 1.1 nM), RON (IC₅₀ = 1.8 nM) and c-Met (IC₅₀ = 3.9 nM). Regardless, we found that BMS 777607 treatment phenocopies the TYRO3 depletion spindle positioning phenotype, compatible with the notion that TYRO3 kinase activity is required for function in this process. These new results are shown in Supplemental Figure 4d and mentioned on p. 11 of the revised manuscript.

6) Are TYRO3 ligands important for the observed effect?

> This is certainly an interesting question, but one that appears to fall outside the scope of the present manuscript. Nevertheless, we now highlight this as an important avenue for future investigations (p. 15).

7) Protein expression (and if possible localization) should be validated more carefully. For protein expression, immunoblotting should be used (to check for bands of predicted molecular weight). As indicated by the authors, centrosomal (and mid-body) localization is often an artefact.

> We may be misunderstanding the suggestion of the reviewer, but immunoblotting data was already reported in the original submission for TYRO3 and GAK, including upon their

depletion by siRNAs and targeting by CRISPR/Cas9 (see current Supplemental Figure 3k,l). Likewise, the distribution of TYRO3 and GAK in control cells and in cells depleted of these components is shown in Figure 3a-e (TYRO3) and Figure 4a,b (GAK).

8) STK17B, as well as TYRO3, have been linked with apoptosis and sensing of apoptotic cells, respectively. Can interaction of cells with apoptotic bodies alter their 'polarity' even when interacting with the L-shaped scaffold?

> The reviewer raises the possibility that the presence of apoptotic bodies could have caused a spindle positioning phenotype in other cells depleted of STK17B and TYRO3. This possibility can be excluded because during the step of manual confirmation cells with neighboring dying cells or with cell remnants in their vicinity were excluded from the analysis. That this was the case is now mentioned explicitly in the revised Methods section, as well as in the legend of Figure 1c (p. 8, 21).

9) P2RY5 is neither a kinase, phosphatase or metalloprotease. Neither is TAAR2 or CXCR6. G-protein coupled receptors should be included as a category.

> We thank the reviewer for this comment. The siRNA library indeed includes some components (some G-protein coupled receptors and associated proteins) in addition to kinases, phosphatases and metalloproteases. The description has been changed accordingly in the revised manuscript (pp. 7, 14, 19).

10) A bit strange that kinases such as LLGL and aPKC, with previously described roles in spindle asymmetric division, are not detected in the assay.

> LLGL was not part of the siRNA library because it is not a kinase. As for aPKC, the fact that it was not retained as a *bona fide* candidate reflects the stringent cut off that we decided to employ: aPKC depletion resulted in a phenotype in 29% in the first round and 35% in the second round of screening. Since the value in the first round was <30%, aPKC was not picked as a potential candidate. See also response to point 1 of Reviewer 1. We are discussing this point using p55 as an example on page 15 of the revised manuscript.

References

- Berthet C, Aleem E, Coppola V, Tessarollo L, Kaldis P. 2003. Cdk2 knockout mice are viable. *Current biology : CB* 13: 1775-1785.
- D'Cruz PM, Yasumura D, Weir J, Matthes MT, Abderrahim H, LaVail MM, Vollrath D. 2000. Mutation of the receptor tyrosine kinase gene *Mertk* in the retinal dystrophic RCS rat. *Human molecular genetics* 9: 645-651.
- Kulman JD, Harris JE, Nakazawa N, Ogasawara M, Satake M, Davie EW. 2006. Vitamin K-dependent proteins in *Ciona intestinalis*, a basal chordate lacking a blood coagulation cascade. *Proceedings of the National Academy of Sciences of the United States of America* 103: 15794-15799.
- Lemke G. 2013. Biology of the TAM receptors. *Cold Spring Harbor perspectives in biology* 5: a009076.
- Lemke G, Rothlin CV. 2008. Immunobiology of the TAM receptors. *Nature reviews Immunology* 8: 327-336.
- Li Q, Lu Q, Lu H, Tian S, Lu Q. 2013. Systemic autoimmunity in TAM triple knockout mice causes inflammatory brain damage and cell death. *PLoS one* 8: e64812.

- Linger RM, Keating AK, Earp HS, Graham DK. 2008. TAM receptor tyrosine kinases: biologic functions, signaling, and potential therapeutic targeting in human cancer. *Advances in cancer research* 100: 35-83.
- Machicoane M, de Frutos CA, Fink J, Rocancourt M, Lombardi Y, Garel S, Piel M, Echard A. 2014. SLK-dependent activation of ERMs controls LGN-NuMA localization and spindle orientation. *The Journal of cell biology* 205: 791-799.
- Ortega S, Prieto I, Odajima J, Martin A, Dubus P, Sotillo R, Barbero JL, Malumbres M, Barbacid M. 2003. Cyclin-dependent kinase 2 is essential for meiosis but not for mitotic cell division in mice. *Nature genetics* 35: 25-31.
- Rane SG, Dubus P, Mettus RV, Galbreath EJ, Boden G, Reddy EP, Barbacid M. 1999. Loss of Cdk4 expression causes insulin-deficient diabetes and Cdk4 activation results in beta-islet cell hyperplasia. *Nature genetics* 22: 44-52.
- Rothlin CV, Carrera-Silva EA, Bosurgi L, Ghosh S. 2015. TAM receptor signaling in immune homeostasis. *Annual review of immunology* 33: 355-391.
- Tsou WI, Nguyen KQ, Calarese DA, Garforth SJ, Antes AL, Smirnov SV, Almo SC, Birge RB, Kotenko SV. 2014. Receptor tyrosine kinases, TYRO3, AXL, and MER, demonstrate distinct patterns and complex regulation of ligand-induced activation. *The Journal of biological chemistry* 289: 25750-25763.

Rebuttal Figure 1 legend

- a:** Distribution of angles of cell division axis (α) at metaphase (mean \pm SD of two independent experiments; n > 100 cells in each case) for ctrl (black), LGN (green) and SLK siRNAs (orange). Mann Whitney U statistic was used to compare distributions. * p = 0.0227 (LGN vs. ctrl), p = 0.9290 (SLK vs. ctrl).
- b:** Figure 2C from Machicoane et al. JCB (Machicoane et al. 2014).
- c:** Illustration of STRING pathway analysis of 16 HITs. Connections between molecules are explained below.

Rebuttal Figure 1

REVIEWERS' COMMENTS:

Reviewer #1 (Remarks to the Author):

The authors have made modifications that answer my main concerns, and in particular the rescue of the GAK phenotype by MCAK RNAi strongly reinforces their conclusions.

I still have several minor comments.

p.3, l.40-42: In the second sentence of the introduction, "in animal cell, the position of the mitotic spindle dictates the correct segregation of fate determinants during cell division, as well as the accurate placement of daughter cells within tissues and organisms". Although there are examples of both instances, I find the two parts of the sentence too definitive and a bit misleading, and would suggest a more moderate formulation (eg use "can dictate", "sometimes dictates", or equivalent, instead of "dictates"). There are indeed examples where fate determinants can be segregated independently from spindle orientation, as they are associated with the spindle itself (eg associated with one pole of the spindle); similarly, cell reorganization after division probably contributes as much (and most likely more) than spindle orientation to the accurate placement of daughter cells in tissues.

p.8, l.183-187: I do not think the new hits identified and validated in the present screen can be used in the calculation of the false negative discovery rate. It would only be justified if regulators that are not yet known but did not come out of the screen were also included, but by definition these false negatives are unknown so they cannot be included... So I think the rate can only be calculated on what was known prior to the screen.

p.10, l.233-234: , Fig3e (instead of Fig3d) for TYRO3 siRNA and Fig3d (instead of data not shown) for TYRO3 CRISPR/Cas9 cells

p.11, l.267 and Figure S4g: the alphaTub-GFP signal appears stronger in the picture of TYRO3 the siRNA cell. 1) is it really the case? And if not 2) are imaging conditions similar for the two conditions. Although the quantitation in S4h confirms an increase of cortical PLCdelta staining, the image is not very convincing because the tub-GFP is also much stronger

There are several "data not shown" throughout the manuscript, some of which might be informative for the reader (eg localization of GAK in interphase cells, distribution of NuMA, Dynein or LGN as well as reduction of central spindle microtubules upon GAK depletion, all p.12, microtubule asters following TYRO3 depletion p.15). Why not include them in the supplementary figures?

p.27, first line: "two representative 3 x 0.5nm", I suppose the authors mean μm .

Reviewer #2 (Remarks to the Author):

The authors have satisfactorily addressed all my comments and substantially improved their manuscript.

Reviewer #3 (Remarks to the Author):

I acknowledge the additional work and the sincere attempt to address my (and the other Reviewers') concerns. I believe that addressing some of the pitfalls and apparent paradoxes have improved the manuscript. I am convinced that the work itself is of high quality and performed with significant rigor. While I am still not entirely convinced that, in particular, the TYRO3 function

identified provides sufficient mechanistic understanding and is of in vivo significance, I recognize that the investigators have done due diligence within the constraints of the system being tested. I hope that this discovery spurs new research in this area and an even more improved understanding of this novel biology of TYRO3. Thus, I have no further reservations regarding this manuscript. At this time, I would only like to request the authors to discuss and cite, if they think it is appropriate, the interesting literature on TYRO3 and AXL function in matrix rigidity sensing, cell stiffness regulation and polarity (Masha Prager-Khoutorsky et al., NCB, 2011; Yang et al., Nano Lett, 2016), in case, matrix-rigidity and cell polarization can be extrapolated to spindle orientation.

REVIEWERS' COMMENTS:

We are grateful to the reviewers for their positive assessment of the revised manuscript. We have addressed the remaining minor concerns in full as delineated point-by-point below.

Reviewer #1 (Remarks to the Author):

The authors have made modifications that answer my main concerns, and in particular the rescue of the GAK phenotype by MCAK RNAi strongly reinforces their conclusions.

I still have several minor comments.

A *p.3, l.40-42: In the second sentence of the introduction, “in animal cell, the position of the mitotic spindle dictates the correct segregation of fate determinants during cell division, as well as the accurate placement of daughter cells within tissues and organisms”. Although there are examples of both instances, I find the two parts of the sentence too definitive and a bit misleading, and would suggest a more moderate formulation (eg use “can dictate”, “sometimes dictates”, or equivalent, instead of “dictates”). There are indeed examples where fate determinants can be segregated independently from spindle orientation, as they are associated with the spindle itself (eg associated with one pole of the spindle); similarly, cell reorganization after division probably contributes as much (and most likely more) than spindle orientation to the accurate placement of daughter cells in tissues.*

We agree and changed the wording to “can dictate” (page 3 of the revised manuscript).

B *p.8, l.183-187: I do not think the new hits identified and validated in the present screen can be used in the calculation of the false negative discovery rate. It would only be justified if regulators that are not yet known but did not come out of the screen were also included, but by definition these false negatives are unknown so they cannot be included... So I think the rate can only be calculated on what was known prior to the screen.*

We thank the reviewer for raising this important point. We fully concur and have excluded the hits from this screen in the estimation of the false negative discovery rate, which is now 50% (page 8).

C *p.10, l.233-234: Fig3e (instead of Fig3d) for TYRO3 siRNA and Fig3d (instead of data not shown) for TYRO3 CRISPR/Cas9 cells.*

Apologies about this mix-up; we made the necessary change (page 10, now Figure 4d and Figure 4e). See also point 5 in the response to the editorial comments and requests.

D *p.11, l.267 and Figure S4g: the alphaTub-GFP signal appears stronger in the picture of TYRO3 the siRNA cell. 1) is it really the case? And if not 2) are imaging conditions similar for the two conditions. Although the quantitation in S4h confirms an increase of cortical PLCdelta staining, the image is not very convincing because the tub-GFP is also much stronger.*

We confirm that the imaging settings were identical for all tested conditions and that Fiji settings were the same as well. We now mention this fact explicitly (page 25). In addition, it should be noted that expression levels of EGFP:: α -tubulin varies between cells. Such variability is now also mentioned explicitly (page 25). Moreover, we changed the figure panel to one that is more representative of the data set.

E *There are several “data not shown” throughout the manuscript, some of which might be informative for the reader (eg localization of GAK in interphase cells, distribution of NuMA, Dynein or LGN as well as reduction of central spindle microtubules upon GAK depletion, all p.12, microtubule asters following TYRO3 depletion p.15). Why not include them in the supplementary figures?*

We thank the reviewer for this remark, which echoes that of the editorial team.

There were five such occurrences in the manuscript, which were handled as detailed hereafter, where we also copy and paste in each case the original text for reference.

- Page 10

By contrast, the centrosomal signal also detected by these antibodies in control cells does not appear to be specific since it remains present to a substantial extent upon TYRO3 siRNA (Fig. 3d, Fig. S3o) and in TYRO3 CRISPR/Cas9 cells (data not shown).

The data was in fact already included in Figure 3d, but not referenced. See also points C and E in the response to the comments made by the referees.

The sentence was changed to:

By contrast, the centrosomal signal also detected by these antibodies in control cells does not appear to be specific since it remains present to a substantial extent in TYRO3 CRISPR/Cas9 cells (Fig. 4d) or upon TYRO3 siRNA (Fig. 4e, Supplementary Figure 4n).

- Page 11

We did not detect a centrosomal localization in interphase cells (data not shown), suggestive of a transient association of GAK with spindle poles.

We performed immunofluorescence analysis with antibodies directed against the centriolar protein CP110 in cells expressing myc-tagged full lengths GAK. We now show representative examples of metaphase and interphase cells in the new Supplemental Figure 6a, with the text having been amended accordingly.

- Page 12

We found no difference upon GAK depletion in the distribution of NuMA, Dynein or LGN when compared to control cells (data not shown).

We removed the mention of LGN, for which publication quality data was no longer available, but now report the distribution of NuMA in fixed specimens and DHC::GFP in live cells undergoing mitosis. The text has been altered accordingly to the following:

We found no striking difference upon GAK depletion in the distribution of NuMA or DHC::GFP when compared to control cells (Supplemental Figure 6b,c).

- Page 12

Cells with severe GAK depletion exhibited an apparent reduction of central spindle microtubules, as previously reported⁴⁴ (data not shown).

The data is now shown in the new Supplemental Figure 6f, with the text having been amended accordingly.

- Page 14

We verified the phenotype using different siRNAs and CRISPR/Cas9-mediated impairment for the two strongest hits, TYRO3 and GAK, as well as with different siRNAs for the two weakest hits, STK17B and DAPK1 (data not shown).

We removed the mention of STK17B and DAPK1 to air on the side of caution, because the rescue dataset is not complete for these two candidates.

F p.27, first line: “two representative 3 x 0.5nm”, I suppose the authors mean μ m.

Indeed, sorry about this typo, which has been fixed.

Reviewer #2 (Remarks to the Author):

The authors have satisfactorily addressed all my comments and substantially improved their manuscript.

Reviewer #3 (Remarks to the Author):

G *I acknowledge the additional work and the sincere attempt to address my (and the other Reviewers') concerns. I believe that addressing some of the pitfalls and apparent paradoxes have improved the manuscript. I am convinced that the work itself is of high quality and performed with significant rigor. While I am still not entirely convinced that, in particular, the TYRO3 function identified provides sufficient mechanistic understanding and is of in vivo significance, I recognize that the investigators have done due diligence within the constraints of the system being tested. I hope that this discovery spurs new research in this area and an even more improved understanding of this novel biology of TYRO3. Thus, I have no further reservations regarding this manuscript. At this time, I would only like to request the authors to discuss and cite, if they think it is appropriate, the interesting literature on TYRO3 and AXL function in matrix rigidity sensing, cell stiffness regulation and polarity (Masha Prager-Khoutorsky et al., NCB, 2011; Yang et al., Nano Lett, 2016), in case, matrix-rigidity and cell polarization can be extrapolated to spindle orientation.*

We thank the reviewer for this interesting suggestion, which we have followed in modifying the relevant Discussion section (page 15/16).